# Unified Universality Theorem for Deep and Shallow Joint-Group-Equivariant Machines

## Abstract

We present a constructive universal approximation theorem for learning machines equipped with joint-group-equivariant feature maps, based on the group representation theory. "Constructive" here indicates that the distribution of parameters is given in a closed-form expression known as the ridgelet transform. Joint-group-equivariance encompasses a broad class of feature maps that generalize classical group-equivariance. Notably, this class includes fully-connected networks, which are *not* group-equivariant *but* are joint-group-equivariant. Moreover, our main theorem also unifies the universal approximation theorems for both shallow and deep networks. While the universality of shallow networks has been investigated in a unified manner by the ridgelet transform, the universality of deep networks has been investigated in a case-by-case manner.

## 1 Introduction

The proof of a universality theorem contains hints for understanding the internal data processing mechanisms inside learning machines such as neural networks. For example, the first universality theorem*s* for depth-2 neural networks were shown in 1989 with *four* different proofs by Cybenko (1989), Hornik et al. (1989), Funahashi (1989), and Carroll & Dickinson (1989). Among them, Cybenko's proof using Hahn-Banach and Hornik et al.'s proof using Stone-Weierstrass are existential proofs, meaning that it is not clear how to assign the parameters. On the other hand, Funahashi's proof reducing to the Fourier transform and Carroll and Dickinson's proof reducing to the Radon transform are constructive proofs, meaning that it is clear how to assign the parameters. The latter constructive methods, which reduce to integral transforms, were refined as the so-called integral representation by Barron (1993) and further culminated as the *ridgelet transform*, the main objective of this study, discovered by Murata (1996) and Candès (1998).

To show the universality in a constructive manner, we formulate the the problem as a functional equation: Let $\mathtt{LM}[\gamma]$ denote a certain learning machine (such as a deep network) with parameter $\gamma$, and let $\mathcal{F}$ denote a class of functions to be expressed by the learning machine. Given a function $f \in \mathcal{F}$, find an unknown parameter $\gamma$ so that the machine $\mathtt{LM}[\gamma]$ represents function $f$, i.e.

$$\mathtt{LM}[\gamma] = f, \qquad (1)$$

which we call a *learning equation*. This equation is understood as a stronger formulation of learning than an ordinary formulation by the empirical risk minimization such as minimizing $\sum_{i=1}^{n} |\mathtt{LM}[\gamma](x_i) - f(x_i)|^2$ with respect to $\gamma$, as the latter is understood as a weak form (or a variational form) of this equation. Therefore, characterizing the solution space of this equation leads to understanding the parameters obtained by risk minimization. Following previous studies (Murata, 1996; Candès, 1998; Sonoda et al., 2021a;b; 2022a;b), we call a solution operator $\mathtt{R}$ that satisfies $\mathtt{LM}[\mathtt{R}[f]] = f$ a *ridgelet transform*. Once such a solution operator $\mathtt{R}$ is found, we can conclude a *universality* of the learning machine in consideration because the reconstruction formula $\mathtt{LM}[\mathtt{R}[f]] = f$ implies for any $f \in \mathcal{F}$ there exists a machine that represents $f$. In particular, when $\mathtt{R}[f]$ is found in a closed-form manner, then it leads to a *constructive* proof of the universality since $\mathtt{R}[f]$ could indicate how to assign parameters.

For depth-2 neural networks (particularly with an infinitely-wide hidden layer), the equation has been solved with several closed-form ridgelet transforms. For example, the closed-form ridgelet transforms have been obtained for depth-2 fully-connected layers (Sonoda et al., 2021b), depth-2

fully-connected layers on manifolds (Sonoda et al., 2022b), depth-2 group convolution layers (Sonoda et al., 2022a), and depth-2 fully-connected layers on finite fields (Yamasaki et al., 2023). The essential technique to obtain these ridgelet transforms are to construct a Fourier expression corresponding to the network in consideration. We refer to Sonoda et al. (2024b) for more tecnnical backgrounds behind these results. Furthermore, Sonoda et al. (2021a) have revealed that the distribution of parameters inside depth-2 fully-connected networks obtained by regularized empirical risk minimization assymptotically converges to the ridgelet transform. In other words, the ridgelet transform can also explain the solutions obtained by risk minimization.

On the other hand, for depth-$n$ neural networks, the equation is far from solved, and it is common to either consider infinitely-deep mathematical models such as Neural ODEs (Sonoda & Murata, 2017b; E, 2017; Li & Hao, 2018; Haber & Ruthotto, 2017; Chen et al., 2018), or handcraft solutions depending on the network specifications. For example, construction methods such as the so-called Telgarsky sawtooth function (or the Yarotsky scheme) and bit extraction techniques (Cohen et al., 2016; Telgarsky, 2016; Yarotsky, 2017; 2018; Yarotsky & Zhevnerchuk, 2020; Daubechies et al., 2022; Cohen et al., 2022; Siegel, 2023; Petrova & Wojtaszczyk, 2023; Grohs et al., 2023) have been developed (not only to investigate the expressivity but also) to demonstrate the depth separation, super-convergence, and minmax optimality of deep ReLU networks. Various feature maps have also been handcrafted in the contexts of geometric deep learning (Bronstein et al., 2021) and deep narrow networks (Lu et al., 2017; Hanin & Sellke, 2017; Lin & Jegelka, 2018; Kidger & Lyons, 2020; Park et al., 2021; Li et al., 2023; Cai, 2023; Kim et al., 2024). However, for the purpose of understanding the parameters obtained by risk minimization (in a manner presented by Sonoda et al. (2021a)), these results are less satisfactory because there is no guarantee that these handcrafted solutions are obtaiend by risk minimization.

Recently, Sonoda et al. (2024a) developed a novel technique to show the universality based on the group representation theory, and discovered a rich class of ridgelet transforms for learning machines with joint-group-*invariant* feature maps. However, their technique was essentially limited to depth-2 networks and could not cover depth-$n$ networks defined by composites of nonlinear activation functions such as $\sigma(A_2\sigma(A_1\boldsymbol{x} - \boldsymbol{b}_1) - \boldsymbol{b}_2)$. By carefully reviewing their group theoretic arguments, we found that the joint-*invariance* is the bottleneck and it can be resolved by relaxing the assumption to the joint-*equivariance*. In this study, we present a wider class of ridgelet transforms for learning machines with joint-*equivariant* feature maps so to cover the depth-$n$ (as well as depth-2) fully-connected networks.

The contributions of this study include

- We derived the ridgelet transform (solution operator for the learning equation) for learning machines with depth-$n$ joint-group-equivariant feature maps. Since the solution of the learning equation can be written in closed form for any $f \in \mathcal{F}$, it is a constructive and unified proof of the universal approximation theorem for joint-group-equivariant machines.

- As a corollary, we have shown the constructive universal approximation property of deep fully-connected neural networks. Until this study, the universality of deep networks has been shown in a different manner from the universality of shallow networks, but our results discuss them on common ground. Now we can understand the approximation schemes of various learning machines in a unified manner.

- In addition, as an example of a learning machine whose universality has not been known, we presented a network with quadratic forms and showed its universality.

- Further, we have shown the ridgelet transform for depth-$n$ group convolutional networks. In a previous study, it was known only for depth-2, and we have succeeded to extend it by reviewing the arguments from the group theoretic perspective.

## 2 PRELIMINARIES

We quickly overview the original integral representation and the ridgelet transform, a mathematical model of depth-2 fully-connected network and its right inverse. Then, we list a few facts in the group representation theory. In particular, *Schur's lemma* and the *Haar measure* play key roles in the proof of the main results.

**Notation.** For any topological space $X$, $C_c(X)$ denotes the Banach space of all compactly supported continuous functions on $X$. For any measure space $X$, $L^p(X)$ denotes the Banach space of all $p$-integrable functions on $X$. $\mathcal{S}(\mathbb{R}^d)$ and $\mathcal{S}'(\mathbb{R}^d)$ denote the classes of rapidly decreasing functions (or Schwartz test functions) and tempered distributions on $\mathbb{R}^d$, respectively.

## 2.1 Integral Representation and Ridgelet Transform for Depth-2 Fully-Connected Network

**Definition 1.** For any measurable functions $\sigma : \mathbb{R} \to \mathbb{C}$ and $\gamma : \mathbb{R}^m \times \mathbb{R} \to \mathbb{C}$, put

$$S_\sigma[\gamma](\boldsymbol{x}) := \int_{\mathbb{R}^m \times \mathbb{R}} \gamma(\boldsymbol{a}, b)\sigma(\boldsymbol{a} \cdot \boldsymbol{x} - b)\mathrm{d}\boldsymbol{a}\mathrm{d}b, \quad \boldsymbol{x} \in \mathbb{R}^m. \tag{2}$$

We call $S_\sigma[\gamma]$ an (integral representation of) neural network, and $\gamma$ a parameter distribution.

The integration over all the hidden parameters $(\boldsymbol{a}, b) \in \mathbb{R}^m \times \mathbb{R}$ means all the neurons $\{\boldsymbol{x} \mapsto \sigma(\boldsymbol{a} \cdot \boldsymbol{x} - b) \mid (\boldsymbol{a}, b) \in \mathbb{R}^m \times \mathbb{R}\}$ are summed (or integrated, to be precise) with weight $\gamma$, hence formally $S_\sigma[\gamma]$ is understood as a continuous neural network with a single hidden layer. We note, however, when $\gamma$ is a finite sum of point measures such as $\gamma_p = \sum_{i=1}^p c_i \delta_{(\boldsymbol{a}_i, b_i)}$ (by appropriately extending the class of $\gamma$ to Borel measures), then it can also reproduce a finite width network

$$S_\sigma[\gamma_p](\boldsymbol{x}) = \sum_{i=1}^p c_i \sigma(\boldsymbol{a}_i \cdot \boldsymbol{x} - b_i). \tag{3}$$

In other words, the integral representation is a mathematical model of depth-2 network with *any* width (ranging from finite to continuous).

Next, we introduce the ridgelet transform, which is known to be a right-inverse operator to $S_\sigma$.

**Definition 2.** For any measurable functions $\rho : \mathbb{R} \to \mathbb{C}$ and $f : \mathbb{R}^m \to \mathbb{C}$, put

$$R_\rho[f](\boldsymbol{a}, b) := \int_{\mathbb{R}^m} f(\boldsymbol{x})\overline{\rho(\boldsymbol{a} \cdot \boldsymbol{x} - b)}\mathrm{d}\boldsymbol{x}, \quad (\boldsymbol{a}, b) \in \mathbb{R}^m \times \mathbb{R}. \tag{4}$$

We call $R_\rho$ a ridgelet transform.

To be precise, it satisfies the following reconstruction formula.

**Theorem 1** (Reconstruction Formula). *Suppose $\sigma$ and $\rho$ are a tempered distribution ($\mathcal{S}'$) and a rapid decreasing function ($\mathcal{S}$) respectively. There exists a bilinear form $((\sigma, \rho))$ such that*

$$S_\sigma \circ R_\rho[f] = ((\sigma, \rho))f, \tag{5}$$

*for any square integrable function $f \in L^2(\mathbb{R}^m)$. Further, the bilinear form is given by $((\sigma, \rho)) = \int_{\mathbb{R}} \sigma^\sharp(\omega)\overline{\rho^\sharp(\omega)}|\omega|^{-m}\mathrm{d}\omega$, where $\sharp$ denotes the 1-dimensional Fourier transform.*

See Sonoda et al. (2021b, Theorem 6) for the proof. In particular, according to Sonoda et al. (2021b, Lemma 9), for any activation function $\sigma$, there always exists $\rho$ satisfying $((\sigma, \rho)) = 1$. Here, $\sigma$ being a tempered distribution means that typical activation functions are covered such as ReLU, step function, $\tanh$, gaussian, etc... We can interpret the reconstruction formula as a universality theorem of continuous neural networks, since for any given data generating function $f$, a network with output weight $\gamma_f = R_\rho[f]$ reproduces $f$ (up to factor $((\sigma, \rho))$), i.e. $S[\gamma_f] = f$. In other words, the ridgelet transform indicates how the network parameters should be organized so that the network represents an individual function $f$.

The original ridgelet transform was discovered by Murata (1996) and Candès (1998). It is recently extended to a few modern networks by the Fourier slice method (see e.g. Sonoda et al., 2024b). In this study, we present a systematic scheme to find the ridgelet transform for a variety of given network architecture based on the group theoretic arguments.

## 2.2 Irreducible Unitary Representation and Schur's Lemma

In the main theorem, we use *Schur's lemma*, a fundamental theorem from unitary group representation theory. Group representation is a method for investigating properties of an abstract group $G$ by

mapping $G$ to another (much computable) group of invertible linear operators. We refer to Folland (2015) for more details on group representation and harmonic analysis on groups.

In this study, we assume group $G$ to be *locally compact*. This is a sufficient condition for having invariant measures. It is not a strong assumption. For example, any finite group, discrete group, compact group, and finite-dimensional Lie group are locally compact, while an infinite-dimensional Lie group is *not* locally compact.

Let $\mathcal{H}$ be a nonzero Hilbert space, and $\mathcal{U}(\mathcal{H})$ be the group of unitary operators on $\mathcal{H}$. A *unitary representation* $\pi$ of $G$ on $\mathcal{H}$ is a group homomorphism that is continuous with respect to the strong operator topology—that is, a map $\pi : G \to \mathcal{U}(\mathcal{H})$ satisfying $\pi_{gh} = \pi_g \pi_h$ and $\pi_{g^{-1}} = \pi_g^{-1}$, and for any $\psi \in \mathcal{H}$, the map $G \ni g \mapsto \pi_g[\psi] \in \mathcal{H}$ is continuous.

Suppose $\mathcal{M}$ is a closed subspace of $\mathcal{H}$. $\mathcal{M}$ is called an *invariant* subspace when $\pi_g[\mathcal{M}] \subset \mathcal{M}$ for all $g \in G$. Particularly, $\pi$ is called *irreducible* when it does not admit any nontrivial invariant subspace $\mathcal{M} \neq \{0\}$ nor $\mathcal{H}$. The following theorem is a fundamental result of group representation theory that characterizes the irreducibility.

**Theorem 2** (Schur's lemma). *A unitary representation $\pi : G \to \mathcal{U}(\mathcal{H})$ is irreducible iff any bounded operator $T$ on $\mathcal{H}$ that commutes with $\pi$ is always a constant multiple of the identity. In other words, if $\pi_g \circ T = T \circ \pi_g$ for all $g \in G$, then $T = c\,\mathrm{Id}_{\mathcal{H}}$ for some $c \in \mathbb{C}$.*

See Folland (2015, Theorem 3.5(a)) for the proof. We use this as a key step in the proof of our main theorem.

As a concrete example of an irreducible representation, we use the following regular representation of the affine group $\mathrm{Aff}(m)$ on $L^2(\mathbb{R}^m)$.

**Theorem 3.** *Let $G := \mathrm{Aff}(m) := GL(m) \ltimes \mathbb{R}^m$ be the affine group acting on $X = \mathbb{R}^m$ by $(L, \boldsymbol{t}) \cdot \boldsymbol{x} = L\boldsymbol{x} + \boldsymbol{t}$, and let $\mathcal{H} := L^2(\mathbb{R}^m)$ be the Hilbert space of square-integrable functions. Let $\pi : \mathrm{Aff}(m) \to \mathcal{U}(L^2(\mathbb{R}^m))$ be the regular representation of the affine group $\mathrm{Aff}(m)$ on $L^2(\mathbb{R}^m)$, namely $\pi_g[f](\boldsymbol{x}) := |\det L|^{-1/2} f(L^{-1}(\boldsymbol{x} - \boldsymbol{t}))$ for any $g = (L, \boldsymbol{t}) \in G$. Then $\pi$ is irreducible.*

See Folland (2015, Theorem 6.42) for the proof.

## 2.3 Calculus on Locally Compact Group

By Haar's theorem, if $G$ is a locally compact group, then there uniquely exist left and right invariant measures $\mathrm{d}_l g$ and $\mathrm{d}_r g$, satisfying for any $s \in G$ and $f \in C_c(G)$,

$$\int_G f(sg)\mathrm{d}_l g = \int_G f(g)\mathrm{d}_l g, \quad \text{and} \quad \int_G f(gs)\mathrm{d}_r g = \int_G f(g)\mathrm{d}_r g.$$

Let $X$ be a $G$-space with transitive left (resp. right) $G$-action $g \cdot x$ (resp. $x \cdot g$) for any $(g, x) \in G \times X$. Then, we can further induce the left (resp. right) invariant measure $\mathrm{d}_l x$ (resp. $\mathrm{d}_r x$) so that for any $f \in C_c(G)$,

$$\int_X f(x)\mathrm{d}_l x := \int_G f(g \cdot o)\mathrm{d}_l g, \quad \text{resp.} \quad \int_X f(x)\mathrm{d}_r x := \int_G f(o \cdot g)\mathrm{d}_r g,$$

where $o \in X$ is a fixed point called the origin.

## 3 Main Results

We introduce unitary representations $\pi$ and $\widehat{\pi}$, a *joint-equivariant feature map* $\phi : X \times \Xi \to Y$, a *joint-equivariant machine* $\mathrm{LM}[\gamma; \phi] : X \to Y$, and present the ridgelet transform $\mathrm{R}[f; \psi] : \Xi \to \mathbb{C}$ for joint-equivariant machines, yielding the universality $\mathrm{LM}[\mathrm{R}[f; \psi]; \phi] = c_{\phi,\psi} f$. We note that $\pi$ plays a key role in the main theorem, and the joint-equivariance is an essential property of depth-$n$ fully-connected network.

Let $G$ be a locally compact group equipped with a left invariant measure $\mathrm{d}g$. Let $X$ and $\Xi$ be $G$-spaces equipped with $G$-invariant measures $\mathrm{d}x$ and $\mathrm{d}\xi$, called the *data domain* and the *parameter domain*. respectively. Let $Y$ be a separable Hilbert space, called the *output domain*. Let $\mathcal{U}(Y)$ be

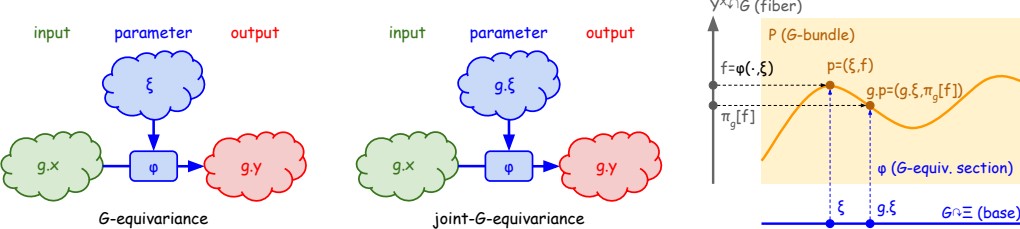

Figure 1: The classical $G$-equivariant feature map $\phi : X \times \Xi \to Y$ is a subclass of joint-$G$-equivariant map where the $G$-action on parameter domain $\Xi$ is *trivial*, i.e. $g \cdot \xi = \xi$

Figure 2: Joint-equivariant map $\phi$ is a $G$-equivariant section of $G$-bundle over base $\Xi$ with fiber $Y^X$

the space of unitary operators on $Y$, and let $\upsilon : G \to \mathcal{U}(Y)$ be a unitary representation of $G$ on $Y$. We call a $Y$-valued map $\phi$ on the data-parameter domain $X \times \Xi$, i.e. $\phi : X \times \Xi \to Y$, a *feature map*.

Let $L^2(X;Y)$ denote the space of $Y$-valued square-integrable functions on $X$ equipped with the inner product $\langle \phi, \psi \rangle_{L^2(X;Y)} := \int_X \langle \phi(x), \psi(x) \rangle_Y \mathrm{d}x$; and let $L^2(\Xi)$ denote the space of $\mathbb{C}$-valued square-integrable functions on $\Xi$.

If there is no risk of confusion, we use the same symbol $\cdot$ for the $G$-actions on $X$, $Y$, and $\Xi$ (e.g., $g \cdot x$, $g \cdot y$, and $g \cdot \xi$). On the other hand, to avoid the confusion between $G$-actions on output domain $Y$ and $Y$-valued function $f : X \to Y$, both "$g \cdot f(x)$" and "$\upsilon_g[f(x)]$" (if needed) always imply $G$-action on $Y$, and "$\pi_g[f](x)$" (introduced soon below) for $G$-actions on $f : X \to Y$.

Additionally, we introduce two unitary representations $\pi$ and $\widehat{\pi}$ of $G$ on function spaces $L^2(X;Y)$ and $L^2(\Xi)$ as follows: For each $g \in G$, $f \in L^2(X;Y)$ and $\gamma \in L^2(\Xi)$,

$$\pi_g[f](x) := \upsilon_g[f(g^{-1} \cdot x)] = g \cdot f(g^{-1} \cdot x), \quad x \in X \tag{6}$$

$$\widehat{\pi}_g[\gamma](\xi) := \gamma(g^{-1} \cdot \xi), \quad \xi \in \Xi. \tag{7}$$

In the main theorem, the irreducibility of $\pi$ will be a sufficient condition for the universality. On the other hand, the irreducibility of $\widehat{\pi}$ is not necessary. For those who are less familiar with group representations, we have shown that $\pi$ and $\widehat{\pi}$ are unitary representations in Lemmas 6 and 7.

### 3.1 JOINT-EQUIVARIANT FEATURE MAP

We introduce the joint-group-equivariant feature map, extending the classical notion of group-equivariant feature maps. The major motivation to introduce this is that depth-$n$ fully-connected networks, the main subject of this study, are not equivariant but joint-equivariant.

**Definition 3** (Joint-$G$-Equivariant Feature Map). We say a feature map $\phi : X \times \Xi \to Y$ is *joint-$G$-equivariant* when

$$\phi(g \cdot x, g \cdot \xi) = g \cdot \phi(x, \xi), \quad (x, \xi) \in X \times \Xi, \tag{8}$$

holds for all $g \in G$. Especially, when $G$-action on $Y$ is trivial, i.e. $\phi(g \cdot x, g \cdot \xi) = \phi(x, \xi)$, we say it is *joint-$G$-invariant*.

*Remark* 1 (Relation to classical $G$-equivariance). The joint-$G$-equivariance is not a restriction but an extension of the classical notion of $G$-*equivariance*, i.e. $\phi(g \cdot x, \xi) = g \cdot \phi(x, \xi)$. In fact, $G$-equivariance is a special case of joint-$G$-equivariance where $G$ acts trivially on parameter domain, i.e. $g \cdot \xi = \xi$ (see Figure 1). Thus, all $G$-equivariant maps are automatically joint-$G$-equivariant.

*Remark* 2 (Interpretations of joint-$G$-equivariance). We have two interpretations from algebraic and geometric perspectives. First, from an algebraic perspective, $\phi$ is a homomorphism (or a $G$-map) between $G$-sets from $X \times \Xi$ to $Y$. So we may denote the collection of all joint-$G$-equivariant maps as $\hom_G(X \times \Xi, Y)$. Second, from a more geometric perspective, $\phi$ is a vector-field $\Xi \to Y^X$ with structure group $G$ acting on fiber $Y^X$ by $\pi$. Here we identify $\phi : X \times \Xi \to Y$ with $\phi_c : \Xi \to Y^X$ by the so-called *currying* $\phi_c(\xi) := \phi(\bullet, \xi)$. In other words, $\phi_c$ is a global section of a trivial $G$-bundle $\Xi \times Y^X \to \Xi$. Consequently, we can understand the $G$-action $g \cdot \xi$ on parameter domain (or

base space) $\Xi$ is induced from the coordinate transformation $\pi$ on fiber $Y^X$ so that the section $\phi_c$ is $G$-equivariant (see Figure 2)

$$\pi_g[\phi_c(\xi)](x) = g \cdot \phi(g^{-1} \cdot x, \xi) =: \phi_c(g \cdot \xi)(x). \tag{9}$$

Finally, two aspects are unified as a tensor-hom adjunction: $\hom_G(X \times \Xi, Y) \cong \hom_G(\Xi, Y^X)$.

In the following, we list several construction methods of joint-equivariant maps in Lemmas 1, 2 and 3 (in the next subsection), indicating the richness of the proposed concept. Whereas to construct a (non-joint) $G$-equivariant network, we must carefully and precisely design the network architecture (see, e.g., a textbook of geometric deep learning Bronstein et al., 2021), to construct a joint-$G$-equivariant network, we can easily and systematically obtain the one.

First, we can synthesize a joint-equivariant map from (not equivariant but) *any* map $\phi_0 : X \to Y$.

**Lemma 1.** *Let $X$ and $Y$ be $G$-sets. Fix an arbitrary map $\phi_0 : X \to Y$, and put $\phi(x, g) := \pi_g[\phi_0](x) = g \cdot \phi_0(g^{-1} \cdot x)$ for every $x \in X$ and $g \in G$. Then, $\phi : X \times G \to Y$ is joint-$G$-equivariant.*

*Proof.* For any $g, h \in G$, we have $\phi(g \cdot x, g \cdot h) = (gh) \cdot \phi_0((gh)^{-1} \cdot (g \cdot x)) = g \cdot \phi(x, h)$. □

In general, a $G$-set is understood as a representation of $G$. So, the case of $X = Y = \Xi = G$ with $\phi : G \times G \to G$ is understood as a primitive type of joint-$G$-equivariant maps $\phi : X \times \Xi \to Y$.

The next lemma suggests the compatibility with function compositions, or deep structures.

**Lemma 2** (Depth-$n$ Joint-Equivariant Feature Map $\phi_{1:n}$). *Given a sequence of joint-$G$-equivariant feature maps $\phi_i : X_{i-1} \times \Xi_i \to X_i$ $(i = 1, \ldots, n)$, let $\Xi_{1:n} := \Xi_1 \times \cdots \times \Xi_n$ be the $n$-fold parameter space with the component-wise $G$-action $g \cdot \xi_{1:n} := (g \cdot \xi_1, \ldots, g \cdot \xi_n)$ for each $n$-fold parameters $\xi_{1:n} \in \Xi_{1:n}$, and let $\phi_{1:n} : X_0 \times \Xi_{1:n} \to X_n$ be the depth-$n$ feature map given by*

$$\phi_{1:n}(x, \xi_{1:n}) := \phi_n(\bullet, \xi_n) \circ \cdots \circ \phi_1(x, \xi_1). \tag{10}$$

*Then, $\phi_{1:n}$ is joint-$G$-equivariant.*

In other words, the composition of joint-equivariant maps defines a cascade product of morphisms: $\hom_G(\Xi_2, X_2^{X_1}) \times \hom_G(\Xi_1, X_1^{X_0}) \to \hom_G(\Xi_1 \times \Xi_2, X_2^{X_0})$. See Appendix A.2 for the proof.

## 3.2 JOINT-EQUIVARIANT MACHINE AND RIDGELET TRANSFORM

We further introduce the joint-equivariant machine, extending the integral representation.

**Definition 4** (Joint-Equivariant Machine). Fix an arbitrary joint-equivariant feature map $\phi : X \times \Xi \to Y$. For any scalar-valued measurable function $\gamma : \Xi \to \mathbb{C}$, define a $Y$-valued map on $X$ by

$$\mathtt{LM}[\gamma; \phi](x) := \int_\Xi \gamma(\xi)\phi(x, \xi)\mathrm{d}\xi, \quad x \in X, \tag{11}$$

where the integral is understood as the Bochner integral. We also write $\mathtt{LM}_\phi := \mathtt{LM}[\bullet; \phi]$ for short. If needed, we call the image $\mathtt{LM}[\gamma; \phi] : X \to Y$ a joint-equivariant *machine*, and the integral transform $\mathtt{LM}[\bullet; \phi]$ of $\gamma$ a joint-equivariant *transform*.

The joint-equivariant machine extends the original integral representation. It inherits the concept of integrating all the possible parameters $\xi$ and indirectly select which parameters to use by weighting on them, which *linearize* parametrization by lifting nonlinear parameters $\xi$ to linear parameter $\gamma$.

Recall that the $G$-action on parameter domain $\Xi$ is also linearized by lifting it to $\widehat{\pi}$ on $L^2(\Xi)$. The joint-equivariance of $\phi : \Xi \to Y^X$ is inherited under the linearization to $\mathtt{LM}_\phi : L^2(\Xi) \to L^2(X; Y)$.

**Lemma 3.** *A joint-$G$-equivariant machine $\mathtt{LM}_\phi : L^2(\Xi) \to L^2(X; Y)$ is joint-$G$-equivariant, i.e. $\mathtt{LM}_\phi \in \hom_G(L^2(\Xi), L^2(X; Y))$.*

*Proof.* $\mathtt{LM}_\phi[\widehat{\pi}_g[\gamma]](g \cdot x) = \int_\Xi \gamma(g^{-1} \cdot \xi)\phi(g \cdot x, \xi)\mathrm{d}\xi = \int_\Xi \gamma(\xi)\phi(g \cdot x, g \cdot \xi)\mathrm{d}\xi = g \cdot \mathtt{LM}_\phi[\gamma](x)$. □

**Definition 5** (Ridgelet Transform for Joint-Equivariant Machine). For any joint-equivariant feature map $\psi : X \times \Xi \to Y$ and $Y$-valued Borel measurable function $f$ on $X$, put a scalar-valued map by

$$\mathtt{R}[f; \psi](\xi) := \int_X \langle f(x), \psi(x, \xi) \rangle_Y \mathrm{d}x, \quad \xi \in \Xi. \tag{12}$$

We also write $\mathtt{R}_\psi := \mathtt{R}[\bullet; \psi]$ for short. If there is no risk of confusion, we call both the image $\mathtt{R}[f; \psi] : X \to Y$ and the integral transform $\mathtt{R}[\bullet; \psi]$ of $f$ a ridgelet transform.

Intuitively, it measures the similarity between target function $f$ and feature $\psi(\bullet, \xi)$ at $\xi$. As long as the integrals are convergent, the ridgelet transform is the dual operator of the joint-equivariant transform (with common $\phi$):

$$\langle \gamma, \mathtt{R}[f; \phi] \rangle_{L^2(\Xi)} = \int_{X \times \Xi} \gamma(\xi) \langle \phi(x, \xi), f(x) \rangle_Y \mathrm{d}x \mathrm{d}\xi = \langle \mathtt{LM}[\gamma; \phi], f \rangle_{L^2(X;Y)}. \tag{13}$$

Similarly to the joint-equivariant machine, the ridgelet transform is again joint-$G$-*invariant*. In fact,

$$\mathtt{R}_\psi[\pi_g[f]](g \cdot \xi) = \int_X \langle \upsilon_g[f(g^{-1} \cdot x)], \psi(x, g \cdot \xi) \rangle_Y \mathrm{d}\xi = \int_X \langle f(x), \upsilon_g^*[\psi(g \cdot x, g \cdot \xi)] \rangle_Y \mathrm{d}\xi = \mathtt{R}_\psi[f](\xi).$$

Hence, geometrically, if we regard $\mathtt{LM}_\phi : L^2(\Xi) \to L^2(X; Y)$ a vector field of trivial $G$-bundle $L^2(\Xi) \times L^2(X; Y) \to L^2(\Xi)$, then $\mathtt{R}_\phi : L^2(X; Y) \to L^2(\Xi)$ corresponds to a $G$-connection.

### 3.3 MAIN THEOREM

At last, we state the main theorem, that is, the reconstruction formula for joint-equivariant machines.

**Theorem 4** (Reconstruction Formula). *Assume (1) feature maps $\phi, \psi : X \times \Xi \to Y$ are joint-$G$-equivariant, (2) composite operator $\mathtt{LM}_\phi \circ \mathtt{R}_\psi : L^2(X; Y) \to L^2(X; Y)$ is bounded (i.e., Lipschitz continuous), and (3) the unitary representation $\pi : G \to \mathcal{U}(L^2(X; Y))$ defined in (6) is irreducible. Then, there exists a bilinear form $((\phi, \psi)) \in \mathbb{C}$ (independent of $f$) such that for any $Y$-valued square-integrable function $f \in L^2(X; Y)$,*

$$\mathtt{LM}_\phi \circ \mathtt{R}_\psi[f] = \int_\Xi \left[ \int_X \langle f(x), \psi(x, \xi) \rangle_Y \mathrm{d}x \right] \phi(\bullet, \xi) \mathrm{d}\xi = ((\phi, \psi)) f. \tag{14}$$

In practice, once the irreducibility of $G$-action $\pi$ on $L^2(X; Y)$ is verified, the ridgelet transform $\mathtt{R}_\psi$ becomes a right inverse operator of joint-equivariant transform $\mathtt{LM}_\phi$ as long as $((\phi, \psi)) \neq 0, \infty$. Despite the wide coverage of examples, the proof is brief and simple as follows.

*Proof.* By using the unitarity of representation $\upsilon : G \to \mathcal{U}(Y)$, left-invariance of measure $\mathrm{d}x$, and $G$-equivariance of feature map $\psi$, for all $g \in G$, we have

$$\mathtt{R}_\psi[\pi_g[f]](\xi) = \int_X \langle g \cdot f(g^{-1} \cdot x), \psi(x, \xi) \rangle_Y \mathrm{d}x = \int_X \langle f(x), g^{-1} \cdot \psi(g \cdot x, \xi) \rangle_Y \mathrm{d}x$$

$$= \int_X \langle f(x), \psi(x, g^{-1} \cdot \xi) \rangle_Y \mathrm{d}x = \widehat{\pi}_g[\mathtt{R}_\psi[f]](\xi). \tag{15}$$

Similarly,

$$\mathtt{LM}_\phi[\widehat{\pi}_g[\gamma]](x) = \int_\Xi \gamma(g^{-1} \cdot \xi) \phi(x, \xi) \mathrm{d}\xi = \int_\Xi \gamma(\xi) \phi(x, g \cdot \xi) \mathrm{d}\xi$$

$$= \int_\Xi \gamma(\xi) \left( g \cdot \phi(g^{-1} \cdot x, \xi) \right) \mathrm{d}\xi = \pi_g[\mathtt{LM}_\phi[\gamma]](x). \tag{16}$$

As a consequence, $\mathtt{LM}_\phi \circ \mathtt{R}_\psi : L^2(X; Y) \to L^2(X; Y)$ commutes with $\pi$ as below

$$\mathtt{LM}_\phi \circ \mathtt{R}_\psi \circ \pi_g = \mathtt{LM}_\phi \circ \widehat{\pi}_g \circ \mathtt{R}_\psi = \pi_g \circ \mathtt{LM}_\phi \circ \mathtt{R}_\psi \tag{17}$$

for all $g \in G$. Hence by Schur's lemma (Theorem 2), there exist a constant $C_{\phi, \psi} \in \mathbb{C}$ such that $\mathtt{LM}_\phi \circ \mathtt{R}_\psi = C_{\phi, \psi} \mathrm{Id}_{L^2(X)}$. Since $\mathtt{LM}_\phi \circ \mathtt{R}_\psi$ is bilinear in $\phi$ and $\psi$, $C_{\phi, \psi}$ is bilinear in $\phi$ and $\psi$. $\square$

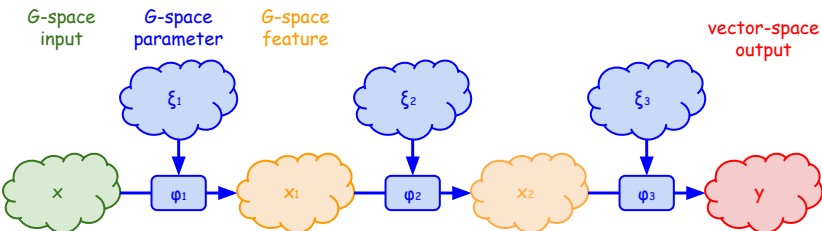

Figure 3: Deep $Y$-valued joint-$G$-equivariant machine on $G$-space $X$ is $L^2(X;Y)$-universal when unitary representation $\pi$ of $G$ on $L^2(X;Y)$ is irreducible, and the distribution of parameters for the machine to represent a given map $f : X \to Y$ is exactly given by the ridgelet transform $\mathtt{R}[f]$

*Remark* 3. (1) When $\pi$ is not irreducible (thus reducible) and admits an irreducible decomposition $L^2(X;Y) = \bigoplus_{i=1}^{\infty} \mathcal{H}_i$, reconstruction formula (14) holds for $f \in \mathcal{H}_k$ for some $k$. This is another consequence from Schur's lemma. (2) The irreducibility is assumed only for $\pi$, and not for $\widehat{\pi}$. This asymmetry originates from the fact that our main theorem focuses only on (the universality of) $\mathtt{LM}_\phi[\gamma] : X \to Y$, not on its dual $\mathtt{R}_\psi[f] : \Xi \to \mathbb{R}$. However, when $\widehat{\pi}$ is irreducible, then we can state $\mathtt{R}_\psi \circ \mathtt{LM}_\phi[\gamma] = \gamma$ for any $\gamma \in L^2(\Xi)$ (the order of composition is reverted from $\mathtt{LM}_\phi \circ \mathtt{R}_\psi$). (3) The regularity of feature maps $\phi, \psi$ needs to be studied in a case-by-case manner. Showing the joint-equivariance is relatively easy. For example, for fully-connected networks (§ 5) and quadratic-form networks (§ 6), the joint-equivariance holds for any activation function. However, the constant $((\phi, \psi))$ can degenerate to zero or diverge depending on feature maps. For the case of depth-2 fully-connected networks, it is known that the constant is zero if and only if the activation function is a polynomial function (see e.g., Sonoda & Murata, 2017a). In general, such a condition can be investigated in a case-by-case manner. Fortunately, we can use the closed-form expression of the ridgelet transform to our advantage.

*Remark* 4 (Technical differences from Sonoda et al. (2024a, Theorem 9)). The previous study cannot deal with deep structures or composite maps and, therefore, falls short as a theoretical analysis of deep learning. This limitation arises because joint-invariance alone is insufficient to construct irreducible representations (or irreps, for short) for deep structures. More technically, it is due to the facts (1) that an *inner* tensor product $\pi_1 \otimes_i \pi_2$ of irreps $\pi_1, \pi_2$ is not always irreducible (see e.g. Chapter 7.3 Folland, 2015), while (2, or Lemma 4) that an *outer* tensor product $\pi_1 \otimes \pi_2$ or irreps $\pi_1, \pi_2$ is always irreducible. The two facts (1) and (2) appear similar (thus confusing) but the essential consequences are different. Further, (3) that deep feature maps are often vector-valued, but (4) that the previous study is limited to scalar-valued joint-invariant feature maps. In the previous study, due to (1) and (4), it is technically hard to obtain an irrep for vector-valued feature maps. Namely, just a $d$-times inner tensor product $\pi_s \otimes_i \cdots \otimes_i \pi_s$ of irrep $\pi_s$ acting on scalar-valued joint-invariant maps $\phi_s$ cannot be an irrep acting on $d$-dim. vector-valued feature map $\phi_s \otimes \cdots \otimes \phi_s$. On the other hand, this study is based on not (1) but (2), we have successfully constructed an irrep as presented in § 5. Furthermore, as demonstrated in Lemmas 1, 2 and 3, the joint-equivariance enables a natural handling of (not only fully-connected but any) deep structures. By seemingly making a small adjustment to the conditions, we were able to address a fundamental problem effectively.

## 4 EXAMPLE: DEPTH-$n$ JOINT-EQUIVARIANT MACHINE

As pointed out in Lemma 2, the depth-$n$ feature map $\phi_{1:n}$ is joint-$G$-equivariant. Therefore, the following $Y$-valued depth-$n$ joint-equivariant machine $\mathtt{DLM}[\gamma; \phi_{1:n}]$ is $L^2(X;Y)$-universal.

**Corollary 1** (Deep Ridgelet Transform). *For any maps $\gamma : X \to \mathbb{C}$ and $f \in L^2(X;Y)$, put*

$$\mathtt{DLM}[\gamma; \phi_{1:n}](x) := \int_{\Xi_1 \times \cdots \times \Xi_n} \gamma(\xi_1, \ldots, \xi_n) \phi_n(\bullet, \xi_n) \circ \cdots \circ \phi_1(x, \xi_1) \mathrm{d}\boldsymbol{\xi}, \quad x \in X, \quad (18)$$

$$\mathtt{R}[f; \psi_{1:n}](\boldsymbol{\xi}) := \int_{\Xi} \langle f(x), \psi_n(\bullet, \xi_n) \circ \cdots \circ \psi_1(x, \xi_n) \rangle_Y \mathrm{d}x, \quad \boldsymbol{\xi} \in \Xi_1 \times \cdots \times \Xi_n. \quad (19)$$

*Under the assumptions that $\mathtt{DLM}_{\phi_{1:n}} \circ \mathtt{R}_{\psi_{1:n}}$ is bounded, and that $\pi$ is irreducible, there exists a bilinear form $((\phi_{1:n}, \psi_{1:n}))$ satisfying $\mathtt{DLM}_{\phi_{1:n}} \circ \mathtt{R}_{\psi_{1:n}} = ((\phi_{1:n}, \psi_{1:n})) \operatorname{Id}_{L^2(X;Y)}$.*

Again, it extends the original integral representation, and inherits the *linearization* trick of nonlinear parameters $\boldsymbol{\xi}$ by integrating all the possible parameters (beyond the difference of layers) and indirectly select which parameters to use by weighting on them.

## 5 EXAMPLE: DEPTH-$n$ FULLY-CONNECTED NETWORK

We explain the case of depth-$n$ (precisely, depth-$n + 1$) fully-connected network. We use the following fact.

**Lemma 4** (Folland (2015, Theorem 7.12)). *Let $\pi_1$ and $\pi_2$ be representations of locally compact groups $G_1$ and $G_2$, and let $\pi_1 \otimes \pi_2$ be their outer tensor product, which is a representation of the product group $G_1 \times G_2$. Then, $\pi_1$ and $\pi_2$ are irreducible if and only if $\pi_1 \otimes \pi_2$ is irreducible.*

Set $X = Y = \mathbb{R}^m$ (input and output domains), and for each $i \in \{1, \ldots, n\}$, set $X_i := \mathbb{R}^{d_i}$ (with $X_1 = X$ and $X_{n+1} = Y$), $\Xi_i := \mathbb{R}^{p_i \times d_i} \times \mathbb{R}^{p_i} \times \mathbb{R}^{d_{i+1} \times q_i}$ (parameter domain), $\sigma_i : \mathbb{R}^{p_i} \to \mathbb{R}^{q_i}$ (activation functions), and define the feature map (vector-valued fully-connected neurons) as

$$\phi_i(\boldsymbol{x}_i, \boldsymbol{\xi}_i) := C_i \sigma_i(A_i \boldsymbol{x}_i - \boldsymbol{b}_i), \quad \boldsymbol{x}_i \in \mathbb{R}^{d_i}, \boldsymbol{\xi}_i = (A_i, \boldsymbol{b}_i, C_i) \in \Xi_i \tag{20}$$

Specifically, $d_1 = d_{n+1} = m$. If there is no risk of confusion, we omit writing $i$ for simplicity.

Let $O(m)$ denote the orthogonal group in dimension $m$. Let $G := O(m) \times \mathrm{Aff}(m)$ be the product group of $O(m)$ and $\mathrm{Aff}(m) = GL(m) \ltimes \mathbb{R}^m$. We suppose $G$ acts on the input and output domains as below: For any $g = (Q, L, \boldsymbol{t}) \in G = O(m) \times (GL(m) \ltimes \mathbb{R}^m)$,

$$g \cdot \boldsymbol{x} := L\boldsymbol{x} + \boldsymbol{t}, \; \boldsymbol{x} \in X, \quad \text{and} \quad g \cdot \boldsymbol{y} := v_g[\boldsymbol{y}] := Q\boldsymbol{y}, \; \boldsymbol{y} \in Y. \tag{21}$$

Namely, the group actions of both $O(m)$ on $X$ and $\mathrm{Aff}(m)$ on $Y$ are trivial.

Let $\pi$ be the induced representation of $G$ on the vector-valued square-integrable functions $L^2(X; Y)$, defined by

$$\pi_g[\boldsymbol{f}](\boldsymbol{x}) := |\det L|^{-1/2} Q \boldsymbol{f}(L^{-1}(\boldsymbol{x} - \boldsymbol{t})), \quad \boldsymbol{x} \in X, \; \boldsymbol{f} \in L^2(X; Y) \tag{22}$$

for each $g = (Q, L, \boldsymbol{t}) \in O(m) \times (GL(m) \ltimes \mathbb{R}^m)$.

**Lemma 5.** *The above $\pi : G \to \mathcal{U}(L^2(\mathbb{R}^m; \mathbb{R}^m))$ is irreducible.*

*Proof.* Recall the representations of $O(m)$ on $\mathbb{R}^m$ and of $\mathrm{Aff}(m)$ on $L^2(\mathbb{R}^m)$ are respectively irreducible (see Theorem 3), and $L^2(\mathbb{R}^m; \mathbb{R}^m)$ is equivalent to the tensor product $\mathbb{R}^m \otimes L^2(\mathbb{R}^m)$. Hence by Lemma 4, the representation $\pi$ of the product group $O(m) \times \mathrm{Aff}(m)$ on the tensor product $\mathbb{R}^m \otimes L^2(\mathbb{R}^m) = L^2(\mathbb{R}^m; \mathbb{R}^m)$ is irreducible. $\square$

Additionally, we put the dual action of $G$ on parameter domain $\Xi_i$ as below:

$$g \cdot (A_i, \boldsymbol{b}_i, C_i) := \begin{cases} (A_i L^{-1}, \boldsymbol{b}_i + A_i L^{-1} \boldsymbol{t}, C_i), & i = 1 \\ (A_i, \boldsymbol{b}_i, C_i), & i \neq 1, n \\ (A_i, \boldsymbol{b}_i, QC_i), & i = n \end{cases} \tag{23}$$

for all $g = (Q, L, \boldsymbol{t}) \in O(m) \times (GL(m) \ltimes \mathbb{R}^m)$, $(A_i, \boldsymbol{b}_i, C_i) \in \Xi_i$.

Then, the composition of feature maps $\phi_{1:n}(\boldsymbol{x}, \boldsymbol{\xi}_{1:n}) := \phi_n(\bullet, \boldsymbol{\xi}_n) \circ \cdots \circ \phi_1(\boldsymbol{x}, \boldsymbol{\xi}_1)$ is joint-$G$-equivariant. In fact,

$$\phi_1(g \cdot \boldsymbol{x}, g \cdot \boldsymbol{\xi}_1) = C_1 \sigma \left( A_1 L^{-1}(L\boldsymbol{x} + \boldsymbol{t}) - (\boldsymbol{b}_1 + A_1 L^{-1} \boldsymbol{t}) \right) = C_1 \sigma(A_1 \boldsymbol{x} - \boldsymbol{b}_1) = \phi_1(\boldsymbol{x}, \boldsymbol{\xi}_1),$$

$$\phi_i(\boldsymbol{x}, g \cdot \boldsymbol{\xi}_i) = C_i \sigma(A_i \boldsymbol{x} - \boldsymbol{b}_i) = \phi_i(\boldsymbol{x}, \boldsymbol{\xi}_i), \quad i \neq 1, n$$

$$\phi_n(\boldsymbol{x}, g \cdot \boldsymbol{\xi}_n) = QC_n \sigma(A_n \boldsymbol{x} - \boldsymbol{b}_n) = g \cdot \phi_n(\boldsymbol{x}, \boldsymbol{\xi}_n),$$

Therefore $\phi_{1:n}(g \cdot \boldsymbol{x}, g \cdot \boldsymbol{\xi}_{1:n}) = g \cdot \phi_{1:n}(\boldsymbol{x}, \boldsymbol{\xi}_{1:n})$.

So by putting depth-$n$ neural network and the corresponding ridgelet transform as below

$$\mathtt{DNN}[\gamma; \phi_{1:n}](\boldsymbol{x}) = \int_{\Xi_{1:n}} \gamma(\boldsymbol{\xi}_{1:n}) \phi_{1:n}(\boldsymbol{x}, \boldsymbol{\xi}_{1:n}) \mathrm{d}\boldsymbol{\xi}_{1:n}, \tag{24}$$

$$\mathtt{R}[\boldsymbol{f}; \psi_{1:n}](\boldsymbol{\xi}_{1:n}) = \int_{\mathbb{R}^m} \boldsymbol{f}(\boldsymbol{x}) \cdot \overline{\psi_{1:n}(\boldsymbol{x}, \boldsymbol{\xi}_{1:n})} \mathrm{d}\boldsymbol{x}, \tag{25}$$

Theorem 4 yields the reconstruction formula $\mathtt{DNN}_{\phi_{1:n}} \circ \mathtt{R}_{\psi_{1:n}} = ((\!(\phi_{1:n}, \psi_{1:n})\!)) \mathrm{Id}_{L^2(\mathbb{R}^m; \mathbb{R}^m)}$.

## 6 EXAMPLE: QUADRATIC-FORM WITH NONLINEARITY

Here, we present a new network for which the universality was not known.

Let $M$ denote the class of all $m \times m$-symmetric matrices equipped with the Lebesgue measure $\mathrm{d}A = \bigwedge_{i \geq j} \mathrm{d}a_{ij}$. Set $X = \mathbb{R}^m$, $\Xi = M \times \mathbb{R}^m \times \mathbb{R}$, and

$$\phi(\boldsymbol{x}, \xi) := \sigma(\boldsymbol{x}^\top A \boldsymbol{x} + \boldsymbol{x}^\top \boldsymbol{b} + c) \tag{26}$$

for any fixed function $\sigma : \mathbb{R} \to \mathbb{R}$. Namely, it is a quadratic-form in $x$ followed by nonlinear activation function $\sigma$.

Then, it is joint-invariant with $G = \mathrm{Aff}(m)$. In fact, we can put the group actions of $g = (\boldsymbol{t}, L) \in \mathbb{R}^m \rtimes GL(m)$ on $X$ and $\Xi$ by

$$(\boldsymbol{t}, L) \cdot \boldsymbol{x} := \boldsymbol{t} + L\boldsymbol{x}, \tag{27}$$

$$(\boldsymbol{t}, L) \cdot (A, \boldsymbol{b}, c) := (L^{-\top} A L^{-1}, L^{-\top} \boldsymbol{b} - 2L^{-\top} A L^{-1} \boldsymbol{t}, c + \boldsymbol{t}^\top L^{-\top} A L^{-1} \boldsymbol{t} - \boldsymbol{t}^\top L^{-\top} \boldsymbol{b}). \tag{28}$$

Then, the joint-$G$-action on $X \times \Xi$ remains the feature map joint-invariant as below.

$$\begin{aligned}
\phi(g \cdot \boldsymbol{x}, g \cdot \boldsymbol{\xi}) &= \sigma((L\boldsymbol{x} + \boldsymbol{t})^\top L^{-\top} A L^{-1} (L\boldsymbol{x} + \boldsymbol{t}) + (L\boldsymbol{x} + \boldsymbol{t})^\top (L^{-\top} \boldsymbol{b} - 2L^{-\top} A L^{-1} \boldsymbol{t}) + ...) \\
&= \sigma(\boldsymbol{x}^\top A \boldsymbol{x} + 2\boldsymbol{x}^\top A L^{-1} \boldsymbol{t} + \boldsymbol{t}^\top L^{-\top} A L^{-1} \boldsymbol{t} + \\
&\quad + \boldsymbol{x}^\top \boldsymbol{b} - 2\boldsymbol{x}^\top A L^{-1} \boldsymbol{t} + \boldsymbol{t}^\top L^{-\top} \boldsymbol{b} - 2\boldsymbol{t}^\top L^{-\top} A L^{-1} \boldsymbol{t} \\
&\quad + c + \boldsymbol{t}^\top L^{-\top} A L^{-1} \boldsymbol{t} - \boldsymbol{t}^\top L^{-\top} \boldsymbol{b}) \\
&= \sigma(\boldsymbol{x}^\top A \boldsymbol{x} + \boldsymbol{x}^\top \boldsymbol{b} + c) = \phi(g \cdot \boldsymbol{x}, g \cdot \boldsymbol{\xi}).
\end{aligned}$$

By Theorem 3, the regular representation $\pi$ of $\mathrm{Aff}(m)$ on $L^2(\mathbb{R}^m)$ is irreducible. Hence as a consequence of the general result, the following network is $L^2(\mathbb{R}^m)$-universal.

$$\mathtt{NN}[\gamma](\boldsymbol{x}) := \int_{M \times \mathbb{R}^m \times \mathbb{R}} \gamma(A, \boldsymbol{b}, c) \sigma(\boldsymbol{x}^\top A \boldsymbol{x} + \boldsymbol{x}^\top \boldsymbol{b} + c) \mathrm{d}A \mathrm{d}\boldsymbol{b} \mathrm{d}c. \tag{29}$$

## 7 DISCUSSION

We have developed a systematic method for deriving a ridgelet transform for a wide range of learning machines defined by joint-group-equivariant feature maps, yielding the universal approximation theorems as corollaries. Traditionally, the techniques used in the expressive power analysis of deep networks were different from those used in the analysis of shallow networks, as overviewed in the introduction. Our main theorem unifies the approximation schemes of both deep and shallow networks from the perspective of joint-group-action on the data-parameter domain. Technically, this unification is due to Schur's lemma, a basic and useful result in the representation theory. Thanks to this lemma, the proof of the main theorem is simple, yet the scope of application is wide. The significance of this study lies in revealing the close relationship between machine learning theory and modern algebra. With this study as a catalyst, we expect a major upgrade to machine learning theory from the perspective of modern algebra.

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

# A PROOFS

## A.1 UNITARITY OF REPRESENTATIONS

**Lemma 6.** $\pi$ *is a unitary representation of* $G$ *on* $L^2(X;Y)$.

*Proof.* Recall that the representation $\upsilon$ of $G$ on $Y$ is unitary. So, for any $g, h \in G$ and $f \in L^2(X;Y)$,

$$\pi_g[\pi_h[f]](x) = g \cdot (h \cdot f(h^{-1} \cdot (g^{-1} \cdot x))) = (gh) \cdot f((gh)^{-1} \cdot x) = \pi_{gh}[f](x),$$

and for any $g \in G$ and $f_1, f_2 \in L^2(X;Y)$,

$$\langle \pi_g[f_1], \pi_g[f_2] \rangle_{L^2(X;Y)} = \int_X \langle \upsilon_g[f_1(g^{-1} \cdot x)], \upsilon_g[f_2(g^{-1} \cdot x)] \rangle_Y \, \mathrm{d}x$$

$$= \int_X \langle f_1(x), \upsilon_g^*[\upsilon_g[f_2(x)]] \rangle_Y \, \mathrm{d}x = \langle f_1, f_2 \rangle_{L^2(X;Y)}. \qquad \square$$

**Lemma 7.** $\widehat{\pi}$ *is a unitary representation of* $G$ *on* $L^2(\Xi)$.

*Proof.* For any $g, h \in G$ and $\gamma \in L^2(\Xi)$,

$$\widehat{\pi}_g[\widehat{\pi}_h[\gamma]](\xi) = \gamma(h^{-1} \cdot (g^{-1} \cdot \xi) = \gamma((gh)^{-1} \cdot \xi) = \widehat{\pi}_{gh}[f](x),$$

and for any $g \in G$ and $\gamma_1, \gamma_2 \in L^2(\Xi)$,

$$\langle \widehat{\pi}_g[\gamma_1], \widehat{\pi}_g[\gamma_2] \rangle_{L^2(\Xi)} = \int_\Xi \gamma_1(g^{-1} \cdot \xi) \overline{\gamma_2(g^{-1} \cdot \xi)} \mathrm{d}\xi$$

$$= \int_\Xi \gamma_1(\xi) \overline{\gamma_2(\xi)} \mathrm{d}\xi = \langle \gamma_1, \gamma_2 \rangle_{L^2(\Xi)}. \qquad \square$$

## A.2 PROOF OF LEMMA 2

*Proof.* For any $g \in G, x \in X$, and $\xi_{1:n} \in \Xi_{1:n}$, we have

$$\phi_{1:n}(g \cdot x, g \cdot \xi_{1:n}) = \phi_n(\bullet, g \cdot \xi_n) \circ \cdots \circ \phi_2(\bullet, g \cdot \xi_2) \circ \phi_1(g \cdot x, g \cdot \xi_1)$$

$$= \phi_n(\bullet, g \cdot \xi_n) \circ \cdots \circ \phi_2(g \cdot \bullet, g \cdot \xi_2) \circ \phi_1(x, \xi_1)$$

$$\vdots$$

$$= \phi_n(g \cdot \bullet, g \cdot \xi_n) \circ \cdots \circ \phi_2(\bullet, \xi_2) \circ \phi_1(x, \xi_1)$$

$$= g \cdot \phi_n(\bullet, \xi_n) \circ \cdots \circ \phi_2(\bullet, \xi_2) \circ \phi_1(x, \xi_1)$$

$$= g \cdot \phi_{1:n}(x, \xi_{1:n}). \qquad \square$$

# B  Example: Depth-$n$ Group Convolutional Network

As mentioned in Remark 1, all the classical equivariant feature maps are automatically joint-equivariant. So, once the irreducibility of representation $\pi$ is verified, our main theorem can state the ridgelet transform for classical equivariant networks. Here, we explain another usage of this study to present the ridgelet transform for *depth-$n$* group convolutional networks (GCNs).

In a previous study (Sonoda et al., 2022a), the authors have introduced a general formulation of *depth-2* GCNs, covering a wide range of typical group equivariant networks such as DeepSets and $\mathrm{E}(n)$-equivariant maps, and presented the ridgelet transform for them in a unified manner. The ridgelet transform for GCNs was derived and shown to satisfy the reconstruction formula based on the *Fourier expression method* (see Sonoda et al., 2024b, for more details), another proof technique for ridgelet transforms that does not require the irreducibility assumption but is limited to depth-2 learning machines.

In the following, we extend the ridgelet transform for GCNs from depth-2 to *depth-$n$* by reviewing it from the group theoretic perspective. The main idea is to turn the depth-$n$ fully-connected network (FCN) $\phi_{1:n}$ in § 5 to a depth-$n$ $G$-convolutional network $\psi_{1:n}$ by following the construction of the previous study.

## B.1  Notations

To begin with, we introduce the affine group $A := \mathrm{Aff}(m) = GL(m) \ltimes \mathbb{R}^m$ as an auxiliary group, besides the primary group $G$ in consideration. Eventually, the irreducibility assumption is required not for $G$ but for $A$. We note that $A$ and $G$ need not satisfy the inclusion relations neither $A \leq G$ nor $G \leq A$. In accordance with the previous study, we write $T_g[\bullet]$ for $G$-action, and write $\alpha \cdot \bullet$ for $A$-action if needed. Caution: Different from § 5, we assign $A$ (instead of $G$) for the affine group acting on $\phi_{1:n}$. So, we will turn a joint-$A$-equivariant map $\phi_{1:n}$ to $G$-equivariant map $\psi_{1:n}$.

Suppose $G$ acts on each intermediate feature space $X_i = \mathbb{R}^{d_i}$ (and acts trivially on each parameter space $\Xi_i$). By abusing notations, we use the same symbols $T$ for every $G$-actions on $X_i$, and $\tau$ for every $G$-actions on function $f : X \to Y$, where $X$ and $Y$ denote any $G$-sets such as $G$ itself and $X_i$, defined by $\tau_g[f](x) := T_g[f(T_{g^{-1}}[x])]$ for every $g, h \in G$. By $L^2_G(X; Y)$, we denote the space of $G$-equivariant $Y$-valued functions $f$ on $X$ that is square-integrable at the identity element $1_G$ of $G$, namely $L^2_G(X; Y) = \{f \in \hom_G(X, Y^G) \mid \|f(\bullet)(1_G)\|_{L^2(X;Y)} < \infty\} \cong \{\tau_\bullet[f_1] \mid f_1 \in L^2(X; Y)\}$.

## B.2  $G$-Convolutional Feature Map

From the $i$-th layer fully-connected map $\phi_i : X_i \times \Xi_i \to X_{i+1}$, we define the $i$-th layer $G$-convolutional map $\psi_i : X_i \times \Xi_i \to X^G_{i+1}$ as follows: For all $\boldsymbol{x}_i \in X_i$ and $\boldsymbol{\xi}_i = (A_i, \boldsymbol{b}_i, C_i) \in \Xi_i$,

$$\psi_i(\boldsymbol{x}_i, \boldsymbol{\xi}_i)(g) := \tau_g[\phi_i](\boldsymbol{x}_i, \boldsymbol{\xi}_i) = T_g[(C_i \sigma_i(A_i T_{g^{-1}}[\boldsymbol{x}_i] - \boldsymbol{b}_i)], \quad g \in G. \tag{30}$$

This is called a $G$-convolutional map because by appropriately specifying the $G$-action $T$, the expression $A_i T_{g^{-1}}[\boldsymbol{x}_i]$ can reproduce a variety of $G$-convolution product, say $a *_T x$, employed in such as DeepSets and $\mathrm{E}(n)$-equivariant maps (see Section 5 of Sonoda et al., 2022a).

Similarly to Lemma 1, each $G$-convolutional map $\psi_i$ is $G$-equivariant in the classical sense (or joint-$G$-equivariant with trivial $G$-action on parameters $\boldsymbol{\xi}_i$) because for any $g, h \in G$,

$$\psi_i(T_g[\boldsymbol{x}_i], \boldsymbol{\xi}_i)(h) = T_h[\phi_i(T_{h^{-1}}[T_g[\boldsymbol{x}_i]], \boldsymbol{\xi}_i)]$$
$$= T_g[T_{g^{-1}h}[\phi_i(T_{(g^{-1}h)^{-1}}[\boldsymbol{x}_i], \boldsymbol{\xi}_i)]] = \tau_g[\psi_i(\boldsymbol{x}_i, \boldsymbol{\xi}_i)](h). \tag{31}$$

We note that the $G$-equivariance holds for any activation function $\sigma_i$, because it is applied in the element-wise manner in $G$.

## B.3  $G$-Convolutional Network and Ridgelet Transform

Next, we define the depth-$n$ $G$-convolutional map $\psi_{1:n} : X \times \Xi_{1:n} \to Y^G$ by their compositions:

$$\psi_{1:n}(\boldsymbol{x}, \boldsymbol{\xi}_{1:n})(g) := \psi_n(\bullet, \boldsymbol{\xi}_n)(g) \circ \cdots \circ \psi_1(\boldsymbol{x}, \boldsymbol{\xi}_1)(g), \tag{32}$$

and define the depth-$n$ $G$-convolutional network by its integration:

$$\mathrm{GCN}[\gamma](\boldsymbol{x})(g) := \int_{\Xi_{1:n}} \gamma(\boldsymbol{\xi}_{1:n})\psi_{1:n}(\boldsymbol{x}, \boldsymbol{\xi}_{1:n})(g)\mathrm{d}\boldsymbol{\xi}_{1:n}. \tag{33}$$

Recall that the depth-$n$ FCN and its ridgelet transform, $\mathrm{R_{fc}}$, are given in (24) and (25) as below.

$$\mathrm{DNN}[\gamma](\boldsymbol{x}) := \int_{\Xi_{1:n}} \gamma(\boldsymbol{\xi}_{1:n})\phi_{1:n}(\boldsymbol{x}, \boldsymbol{\xi}_{1:n})\mathrm{d}\boldsymbol{\xi}_{1:n}, \tag{34}$$

$$\mathrm{R_{fc}}[\boldsymbol{f}](\boldsymbol{\xi}_{1:n}) = \int_{\mathbb{R}^m} \boldsymbol{f}(\boldsymbol{x}) \cdot \overline{\phi'_{1:n}(\boldsymbol{x}, \boldsymbol{\xi}_{1:n})}\mathrm{d}\boldsymbol{x}. \tag{35}$$

Then, as a consequence of Lemmas 2 and 3, we have the following.

**Lemma 8.** $\mathrm{GCN}[\gamma](\boldsymbol{x})(g) = \tau_g[\mathrm{DNN}[\gamma]](\boldsymbol{x}).$

*Proof.* In fact,

$$\begin{aligned}
\psi_{1:n}(\boldsymbol{x}, \boldsymbol{\xi}_{1:n})(g) &= T_g[\phi_n(\bullet, \boldsymbol{\xi}_n) \circ \cdots \circ \phi_1(T_{g^{-1}}[\boldsymbol{x}], \boldsymbol{\xi}_1)] \\
&= T_g[\phi_{1:n}(T_{g^{-1}}[\boldsymbol{x}], \boldsymbol{\xi}_{1:n})] = \tau_g[\phi_{1:n}](\boldsymbol{x}, \boldsymbol{\xi}_{1:n}),
\end{aligned}$$

and thus

$$\mathrm{GCN}[\gamma](\boldsymbol{x})(g) = \int_{\Xi_{1:n}} \gamma(\boldsymbol{\xi}_{1:n})\tau_g[\phi_{1:n}](\boldsymbol{x}, \boldsymbol{\xi}_{1:n})\mathrm{d}\boldsymbol{\xi}_{1:n} = \tau_g[\mathrm{DNN}[\gamma]](\boldsymbol{x}). \qquad \square$$

Finally, the ridgelet transform $\mathrm{R_{conv}}$ for depth-$n$ GCNs is given by using $\mathrm{R_{fc}}$ for depth-$n$ FCNs as below: For any $f \in L^2_G(X : Y)$, put

$$\mathrm{R_{conv}}[f](\boldsymbol{\xi}_{1:n}) := \mathrm{R_{fc}}[f(\bullet)(1_G)](\boldsymbol{\xi}_{1:n}). \tag{36}$$

*Proof.* $\mathrm{GCN}[\mathrm{R_{conv}}[f]](\boldsymbol{x})(g) = \tau_g[\mathrm{DNN}[\mathrm{R_{fc}}[f(\bullet)(1_G)]]](\boldsymbol{x}) = \tau_g[f(\bullet)(1_G)](\boldsymbol{x}) = f(\boldsymbol{x})(g).$ $\qquad \square$

In other words, the ridgelet transform only encodes the information of function $f$ at $1_G$ because the essential information of $f$ is summarized at $1_G$ due to its $G$-equivariance, and the $G$-convolutions in depth-$n$ GCN have a mechanism to automatically expand the summarized information to entire $G$ by using $G$-equivariance. When the depth $n = 2$, it reduces to the ridgelet transform for depth-2 GCNs presented in Theorem 1 of Sonoda et al. (2022a).

We remark that the base feature map $\phi_i$ need not be the fully-connected network. As suggested from the construction at (30), it can be an arbitrary joint-$A$-equivariant map. However, we need to verify the irreducibility of representation $\pi$ of $A$ on $L^2(X; Y)$ in general.

