# OpenReview forum: "Unified Universality Theorem for Deep and Shallow Joint-Group-Equivariant Machines"
_ICLR.cc/2025/Conference — Submitted to ICLR 2025_

### Official Review · Reviewer_8KVT · 2024-10-27

**Soundness:** 3
**Presentation:** 4
**Contribution:** 3
**Rating:** 6
**Confidence:** 3

**Summary:**

The paper presents a new theorem establishing universality of joint group-equivariant learning machines (where a locally compact group acts on input, parameter and output space). The proof is constructive, in that an integral transform is given that maps the function to be represented to a parameter distribution (generalized function). As a special case, this covers universality of deep fully-connected networks, for which such a constructive proof had not been available (it was only known for shallow ones). The approach of this paper applies equally to shallow and deep networks. The paper also introduces new techniques based on group representation theory, that may find wider application in the theory of neural networks.

The basic idea of the proof is to show that the map LM(R(f)) commutes with an irreducible representation and is therefore by Schur's lemma a multiple of the identity. Here R is the operator taking functions to parameters, and LM is the operator mapping parameters to functions.

**Strengths:**

The paper is written in a clear and precise way, and the authors are clearly expert on the topic. Although the result is ultimately a simple application of Schur's lemma, the insight that this tool can be applied here to answer universality questions, and the whole setup that enables the statement of the theorem, is in my opinion quite significant.

**Weaknesses:**

I could not find very significant weaknesses, but one limitation seems to be that although these techniques establish universality for potentially infinitely wide (and finitely deep) neural networks, they do not show that the approximation error can be reduced by increasing width (or at what rate this error reduces).

**Questions:**

In so far as I could tell, the mathematics is correct. I once read Folland's book (a primary reference), so I should have some appropriate background, but still this is a long time ago and with the time I could invest in the paper I could not ascertain with 100% confidence that all results are correct. I am fairly confident in the main theorem, but I found it a bit hard to see when exactly the conditions are satisfied. Seeing more examples of concrete networks where the conditions are satisfied, and importantly, examples where they fail, would help me better appreciate the theorem and convince myself of its significance.

---

> ### Author Response · Authors · 2024-11-18
>
> We thank the reviewer for their high score and positive comments about our paper.
>
> ## Comments on Weaknesses
>
> - … they do not show that the approximation error can be reduced by increasing width (or at what rate this error reduces).
>
> We agree that the approximation rate is an important step to further conduct generalization error analysis. In general, there have already been known several approaches to obtain an approximation rate (such as Barron’s rate and Jackson’s rate), and the focus of this paper may diverge, it would be a subject for future work.
>
> ## Answers to Questions
>
> - … I am fairly confident in the main theorem, but I found it a bit hard to see when exactly the conditions are satisfied. Seeing more examples of concrete networks where the conditions are satisfied, and importantly, examples where they fail, would help me better appreciate the theorem and convince myself of its significance.
>
> As also commented to Reviewer Vqs4, further investigations on the sufficient condition for Theorem 4 are important directions of research in deep learning theory. As presented in Lemmas 1-3, constructing joint-equivariant maps is much easier than constructing classical (non-joint) equivariant maps. In fact, in fully-connected networks (Sec.5) and quadratic-form networks (Sec.6), the joint-equivariance holds for any activation function. It is rather difficult to state the condition when joint-equivariance cannot hold. We note that joint-equivariance does not immediately imply universality, that is, the constant $((\phi,\psi))$ in Theorem 4 could degenerate to zero or diverge depending on feature maps. For the case of depth-2 fully-connected networks, it is known that the constant is zero if and only if the activation function is a polynomial function. In general, such a condition can be investigated in a case-by-case manner. Fortunately, we can use the closed-form expression of the ridgelet transform to our advantage.  (We have supplemented this in Remark 3-3)

---

> ### Comment · Reviewer_8KVT · 2024-11-22
> **Update**
>
> Reviewer oWLq pointed out that there is significant overlap with an earlier paper. I have looked at it and I agree that the overlap is non-trivial, although I still consider the present paper novel enough (as argued also in the author rebuttal). I will update my score to 6.

---

> > ### Author Response · Authors · 2024-11-22
> >
> > Thank you for the update. Let us remind the essential difference from the previous work. That is, as also mentioned in the Introduction, the previous study cannot deal with deep structures such as $\sigma(A_2 \sigma( A_1 x - b_1 ) - b_2)$, and we have resolved this technical limitation.
> >
> > This is a significant contribution to the field of machine learning theory. This is because composite mappings, such as those in neural networks, are inherently challenging to handle not only in ridgelet analysis but also in general functional analysis and harmonic analysis. In fact, most existing deep learning theories have relied on strong assumptions, leading to highly restrictive analyses. For instance, approaches like mean-field theory are limited to depth-2 ReLU networks, and NTK (neural tangent kernel) theories employ perturbative approximations. Other theories impose assumptions such as Gaussian inputs, linear deep networks (=matrices), or continuous infinite-depth models like neural ODEs. These strong approximations create a considerable gap between theory and real-world neural networks. However, due to the lack of suitable mathematical tools, such approaches have been reluctantly accepted as the current standard. Therefore, the outcomes of this study are expected to have a profound impact on the future direction of deep learning theory.

---

> > > ### Author Response · Authors · 2024-11-24
> > >
> > > As the discussion period is closing, we kindly ask you to revise your score. Please keep in mind that the comments from Reviewer oWLq primarily emphasize differences in presentation rather than substantive content.
> > >
> > > In the future, when you propose a new learning machine, the main theorem of this study, along with its concrete examples, will undoubtedly be invaluable in systematically and theoretically demonstrating that the machine possesses sufficient representational power. Without this main theorem, one would likely be left with either constructing a single, highly impractical solution after extensive calculations or reducing the problem to known existence theorems, such as the universality of polynomials or MLPs. If your proposed learning machine can ultimately be reduced to polynomial functions, wouldn’t it be reasonable to replace your machine with polynomial functions instead? By utilizing the main theorem of this study, it becomes possible to derive countless constructive solutions in a relatively systematic manner. While the proof of the main theorem may appear similar, [1] alone is insufficient to address these challenges. Therefore, researchers are likely to prefer citing this study over [1]. We encourage you to carefully consider the practicality and significance of the theorem presented in this work.

---

### Official Review · Reviewer_oWLq · 2024-11-02

**Soundness:** 4
**Presentation:** 2
**Contribution:** 1
**Rating:** 3
**Confidence:** 3

**Summary:**

The proposed paper presents a constructive universal approximation theorem for learning machines equipped with joint-group equivariant feature maps, extending previous work on convolutional and joint-group invariant feature maps. Additionally, it provides examples of depth-$n$ joint-equivariant machines, depth-$n$ fully connected networks, and quadratic forms with nonlinearity.

**Strengths:**

- The paper generalizes existing work on constructive approximation and the ridgelet transform on invariant machines to the equivariant case.
- Mathematical notation is clear, and the proofs are correct.

**Weaknesses:**

1. Significant overlap with previous work [1].
2. Lack of examples covering models commonly used in the equivariant ML literature.
3. Numerous technical lemmas hinder readability and obscure the main message.

Regarding the first weakness, I understand that group equivariance generalizes invariance, extending the scope of this work beyond [1]. However, if I am not mistaken, the technical advancements over [1] to achieve this generalization are minimal. Indeed, the *principal contributions* of this paper—the statement and proof of the main theorem (Theorem 4)—appear *nearly identical* to those of Theorem 9 in [1].

Moreover, it appears that Section 3 (Main Results) of the paper presents only minor differences from Section 3 of [1], which can be summarized as follows:

- Jointly-Equivariant Machines (Def 4) are simply the equivariant analogs of Generalized Neural Networks ([1], Def 7), with no further differences in definition.
- The Ridgelet Transform for Jointly-Equivariant Machines (Def 5) is analogous to the Ridgelet Transform of [1] but is integrated over a scalar product on $Y$ instead of a scalar product on $\mathbb{C}$.
- This section presents four technical lemmas with simple, straightforward proofs that, in my opinion, hinder readability and could be moved to an appendix.

Additionally, Section 2 of the paper and Section 2 of [1] are again nearly identical, though this may be more understandable as they introduce notation.

**Questions:**

- Can the authors clarify the differences between the presented paper and [1] and explain how this work provides novel insights relative to the cited paper?
- Could the authors elaborate on the relevance of the examples presented in Sections 5 and 6 within the context of current equivariant ML literature?

---
[1] S. Sonoda et al., Joint Group Invariant Functions on Data-Parameter Domain Induce Universal Neural Networks. In Proceedings of the 2nd NeurIPS, Workshop on Symmetry and Geometry in Neural Representations, Proceedings of Machine Learning Research, PMLR, 2024.

---

> ### Author Response · Authors · 2024-11-18
>
> We thank the reviewer for their thoughtful comments and questions.
>
> ## Comments on Weaknesses
>
> - Significant overlap with previous work [1].
>
> The mere similarity in formal appearance does not necessarily imply similarity in meaning or content. For example, while the Eulerian path problem can be solved efficiently, the Hamiltonian path problem, which appears quite similar, is NP-complete. Similarly, while a linear combination of affine functions remains affine, a linear combination of affine functions followed by tanh, namely a neural network, can approximate any continuous function.
>
> This study is different from [1] in the class of learning machines. The previous work cannot deal with deep structures or composite maps and, therefore, falls short as a theoretical analysis of deep learning. This limitation arises because joint-invariance alone is insufficient to construct irreducible representations for deep structures.
>
> More technically, it is due to the facts (1) that an **inner** tensor product $\pi_1 \otimes_i \pi_2$ of irreps $\pi_1, \pi_2$ is not always irreducible, while (2, or Lemma 4) that an **outer** tensor product $\pi_1 \otimes \pi_2$ or irreps $\pi_1, \pi_2$ is always irreducible. The two facts (1) and (2) appear similar (thus confusing) but the essential consequences are different. Further, (3) that deep feature maps are often vector-valued, but (4) that the previous study is limited to scalar-valued joint-invariant feature maps. In the previous study, due to (1) and (4), it is technically hard to obtain an irrep for vector-valued feature maps. Namely, just a $d$-times inner tensor product $\pi_s \otimes_i \cdots \otimes_i \pi_s$ of irrep $\pi_s$ acting on scalar-valued joint-invariant maps $\phi_s$ cannot be an irrep acting on $d$-dim. vector-valued feature map $\phi_s \otimes \cdots \otimes \phi_s$. On the other hand, this is based on not (1) but (2), we have successfully constructed an irrep as presented in Section 5 (Depth-n Fully-Connected Network). Furthermore, as demonstrated in Lemmas 1–3, the joint-equivariance enables a natural handling of (not only fully-connected but any) deep structures. By seemingly making a small adjustment to the conditions, we were able to address a fundamental problem effectively. (We have supplemented this in Remark 4)
>
> - Lack of examples covering models commonly used in the equivariant ML literature.
>
> As mentioned in Remark 1, all the equivariant feature maps are automatically joint-equivariant.
> Hence by assuming the irreducibility of an appropriate unitary representation $\pi$ of $G$ on $L^2(X;Y)$, we can conclude the universality from our main theorem. In fact, however, the universality of group-equivariant networks have already been shown in a unified manner by using the ridgelet transform in Sonoda et al. (2022a). The feature map is defined in Eq.19 as $\phi( x, (a,b) )(h) = \sigma( (a *_T x)(h) - b )$, it is indeed joint-equivariant because $\phi( g.x, g.(a,b) )(h) = \sigma( \langle(h^{-1}g).x, a\rangle- b ) = \phi( x, (a,b) )(g^{-1}h)$ with $G$-action on parameters $(a,b)$ being trivial, and as showcased in Section 5, this feature map covers a wide range of typical group-equivariant networks such as Deep Sets and $E(n)$-equivariant maps. We have supplemented this fact in the revised manuscript. (We have supplemented this in Section 7)
>
> - Numerous technical lemmas hinder readability and obscure the main message.
>
> Those Lemmas are stated as concrete examples of joint-equivariant maps. We have moved some proofs to Appendix, and supplemented more comments and remarks.
>
> ## Answers to Questions
>
> - Can the authors clarify the differences between the presented paper and [1] and explain how this work provides novel insights relative to the cited paper?
>
> Please refer to Comments on Weaknesses.
>
> - Could the authors elaborate on the relevance of the examples presented in Sections 5 and 6 within the context of current equivariant ML literature?
>
> As mentioned in the abstract, these are not equivariant networks. Joint-equivariance is not a restriction but an extension of classical non-joint equivariance. Thus our main theorem covers classical equivariant networks. Please also refer to Comments on Weaknesses.

---

> > ### Comment · Reviewer_oWLq · 2024-11-22
> >
> > I thank authors for the extensive answer and the valuable additions to the manuscript which for sure improve clarity and accessibility of the paper.
> >
> > As I highlighted as a strength in my review, the paper extends existing work on constructive approximation and the ridgelet transform for invariant machines to the equivariant case. Furthermore, I recognize that joint-group equivariance generalizes joint-group invariance, broadening the scope of this work beyond [1] and addressing a wider range of relevant models.
> >
> > I feel that your response further clarifies and reinforces this aspect, which I had already evaluated positively in my review.
> >
> > However, as I may not have emphasized sufficiently in the review, my primary concern lies in the striking similarity between the Main Theorem of [1] and the Main Theorem of the submitted paper, which is presented as the primary contribution of the work. In particular, I want to stress that my concern does not refer to the scope of the theorem itself. Rather, it centers on the fact that the ideas and techniques used to prove it are identical to those employed in [1], resulting in a contribution gap between the presented paper and [1] that might be deemed insufficient to justify publication.
> >
> > Given that determining what constitutes sufficient novelty and a substantial contribution is inherently subjective, allow me to illustrate my perspective using a thoughtful example provided by the authors:
> >
> > > The mere similarity in formal appearance does not necessarily imply similarity in meaning or content. [...] while a linear combination of affine functions remains affine, a linear combination of affine functions followed by tanh, namely a neural network, can approximate any continuous function.
> >
> > I agree that the two theories, which diverge due to a simple modification, show significant differences, with the second being far richer than the first. However, the research efforts that were required to study these theories also differ. For example, new tools had to be developed to prove the universality of neural networks, and their development constituted substantial research contributions.
> >
> > However, I still believe that, unlike in this example, the tools presented in the submitted manuscript, particularly the proof of Theorem 1, are largely derivative of those introduced in [1].
> >
> > Please note that I do not intend to oppose to this research direction, as I find it significant and full of potential. However, I advocate for further progress before it is ready for publication.
> >
> > That said, I remain open to discussions with the authors and other reviewers and welcome additional perspectives on this matter.

---

> > > ### Author Response · Authors · 2024-11-22
> > >
> > > The types of techniques used to prove the universality of neural networks are relatively limited. (Beyond the four methods mentioned in the Introduction, there are approaches such as transforming into known universal approximators like B-splines, MLPs, or KANs; and methods like those used for some narrow networks or the Yarotsky scheme, which quantize the input space, permute a finite number of points, and output into continuous space in a three-step process. Most proofs, if read carefully, can be attributed to one of these arguments.) Therefore, expecting a complete overhaul of the proof of the main theorem is an excessively high bar. The property of joint-equivariance itself is a novel characteristic of feature maps that has not been explored before. Additionally, Lemmas 1–3 and Corollary 1 are new theorems as well.
> > >
> > > As for research efforts, it is worth noting that we have spent years in trial and error aiming to innovate the current state of  functional analysis that struggles to handle even a depth-3 network like $\sigma(A_2 \sigma(A_1 x - b_1) -b_2)$ without strong approximations. The sheer volume of research efforts behind this work may not be immediately visible in a single anonymous manuscript. However, in the broader context of over a decade of deep learning theory or even the century-long development of harmonic analysis, we urge you to carefully consider why a contribution described as “the first functional analysis capable of handling depth-$n$ networks without strong approximations” could face rejection.

---

> > > > ### Author Response · Authors · 2024-11-24
> > > >
> > > > We have sincerely addressed all three issues raised by Reviewer oWLq. If there are no further objections, we kindly ask you to revise your score accordingly.
> > > >
> > > > In the future, when reviewers propose novel learning machines and seek to theoretically guarantee their sufficiently high approximation ability (universality), the main theorem of this study will prove highly valuable. This is because it suffices to simply demonstrate **joint-equivariance**. Without this main theorem, however, one would likely be left with either constructing a single, highly unrealistic solution after extensive calculations or reducing the problem to known existence theorems (e.g., the universality of polynomials or MLPs). If your proposed learning machine can be reduced to polynomial functions, does it imply that your proposed machine could be replaced by polynomial functions instead?
> > > >
> > > > By leveraging the main theorem of this study, it is possible to derive countless constructive solutions in a relatively systematic manner. While the proof may appear similar, the previous study [1] is inadequate for networks with deep structures. These capabilities are uniquely enabled by the theorem presented in this study. Therefore, researchers would naturally prefer to cite this study over [1]. Please carefully consider the utility and significance of this theorem.

---

> > > > > ### Comment · Reviewer_oWLq · 2024-11-24
> > > > >
> > > > > I once again thank the authors for engaging in the discussion regarding the points raised in my review and comments. However, for the reasons provided in my comments, I maintain my evaluation of the paper.

---

> > > > ### Comment · Reviewer_oWLq · 2024-11-24
> > > >
> > > > I appreciate the author’s engagement in the discussion and would like to address some of the points they have raised:
> > > >
> > > > > The types of techniques used to prove the universality of neural networks are relatively limited [...]
> > > >
> > > > I agree with the authors on this statement. However, in my view, when a proof closely resembles an existing one, proper attribution to the original source is essential. In such cases, I cannot consider the proof itself to be a significant contribution of the paper.
> > > >
> > > > > Expecting a complete overhaul of the proof of the main theorem is an excessively high bar.
> > > >
> > > > I am not advocating for a complete overhaul of the proof. Since the proof appears correct to me, I see no reason for it to be modified. However, for the reason explained above, I do not consider it a substantial contribution of the paper.
> > > >
> > > > Instead, I am advocating for the inclusion of additional relevant material to strengthen the paper's overall contribution.
> > > >
> > > > As a first thought, an example could involve addressing the last weakness highlighted by Reviewer Vqs4, extending beyond the general observation made in Remark 3.3. This could be achieved by providing concrete, detailed and relevant examples that illustrate cases where $((\phi, \psi))$ degenerates to zero, diverges, or is non-trivial, along with a characterization or an explanation of the key factors that trigger these phenomena.

---

> > > > > ### Author Response · Authors · 2024-11-24
> > > > >
> > > > > > As a first thought, an example could involve addressing the last weakness highlighted by Reviewer Vqs4, extending beyond the general observation made in Remark 3.3. This could be achieved by providing concrete, detailed and relevant examples that illustrate cases where
> > > > >  degenerates to zero, diverges, or is non-trivial, along with a characterization or an explanation of the key factors that trigger these phenomena.
> > > > >
> > > > > That is a **misinterpretation**. Reviewer Vqs4 asked whether activation functions are subject to the same strict constraints as equivariant networks. We answered **no** to this question. Remark 3 was included out of our commitment to technical integrity and as a gesture of helpfulness. It simply requires a mechanical check to determine whether a constant is zero or not. To frame this as if it represents a fundamental weakness is entirely misplaced.

---

> > > > > > ### Comment · Reviewer_oWLq · 2024-11-24
> > > > > >
> > > > > > Please note that my main concern remains the similarity with [1]. The previous comment was merely a suggestion for possible additional material to include, and it was not my intention to frame it as a particular weakness of the paper.
> > > > > >
> > > > > > Furthermore, it was not clear to me that the answer to Reviewer Vqs4 was *no*.
> > > > > >
> > > > > > For reference:
> > > > > >
> > > > > > Extract from review:
> > > > > > > It would be important to state which assumptions need to hold for the non-linearities for the joint group equivariant case and group equivariant case (in which G acts trivially on the parameter domain).
> > > > > >
> > > > > > Extract from the answer:
> > > > > > > [...] nonlinearity can easily break the classical (i.e. non-joint) equivariance. On the other hand, joint-equivariance is more likely to hold.
> > > > > >
> > > > > > To me, it is unclear how this answer addresses what happens in the case where $G$ acts trivially on the parameter domain, and joint-equivariance reduces to classical equivariance, which certain nonlinearities can break.
> > > > > >
> > > > > > However, I reiterate that this aspect does not negatively impact my final score.

---

> ### Author Response · Authors · 2024-11-24
>
> We have sincerely addressed all three points of feedback; however, it seems as though this has had no impact on the scores. Is this truly reflective of a **fair** and constructive discussion?

---

> ### Author Response · Authors · 2024-11-24
>
> Reviewer Vqs4 does not ask
> > what happens in the case where $G$ acts trivially on the parameter domain
>
> If you **do not want** to use any non-trivial group action on the parameters, nonlinearities can easily break the equivariance, so you have to carefully handcraft the network architecture so that the network is group equivariant. On the other hand, if you use non-trivial group action, you can much easily find a $G$-action on parameters, as demonstrated in Sections 5 and 6. For example, you can try finding a further extension of the case of quadratic-form to $\sigma \circ \mathrm{polynomial}$.

---

> ### Author Response · Authors · 2024-11-24
>
> Summary of the discussion
>
> The discussion has revolved around the similarities and differences between our work and previous study [1], particularly concerning the main theorem. Below is a summary of the key points raised:
>
> **Factual Points**:
> - Limitations of [1]: Previous Study [1] is limited in its ability to handle depth-$n$ models, a shortcoming that our study explicitly addresses and resolves.
> - Contributions of this study beyond the main theorem:
>   - The introduction of the novel concept of joint-equivariance.
>   - New theoretical results, including Lemmas 1–3.
>   - Explicit definition of constructive solution operators for general depth-$n$ networks (not restricted to fully connected networks), as presented in Corollary 1.
>   - Concrete examples of irreducible representations for depth-$n$ fully connected networks.
>
> **Reviewer’s Opinion**:
> - Reviewer oWLq appears to place significant emphasis on the similarity in the formal structure of the main theorem between our work and [1], which may contribute to an underestimation of the differences in their substantive contributions.
> - The reviewer acknowledges the general principle that similarity in formal appearance does not necessarily imply similarity in meaning or content. Despite this, they express concerns about the perceived "research effort" and use the visual similarity in the proof of the main theorem to question the overall value of the study.
>
> **Authors’ Response**:
> - While the proof of the main theorem may appear similar to that of [1], the broader contributions of this study are distinct and substantial. Focusing solely on a single aspect (such as the similarity in a part of the proof) risks undervaluing the overall contributions of this research.
> - The issue of "research effort" is subjective and difficult to measure objectively. Using it as a basis for rejection leaves limited room for constructive dialogue.
>
> **Open Question**:
> - The authors have asked the reviewer a specific question that remains unanswered:
>
>   *If the reviewer were to use our main theorem in their future work, would they simply cite [1] and repeatedly fill in the gaps themselves to account for the differences?*

---

> ### Comment · Reviewer_Vqs4 · 2024-11-24
>
> I have to say that I do not appreciate the tone of the authors' response to the remarks made by reviewer oWLq. This should be a constructive discussion and the authors should rather be thankful to the reviewers whose reviews have the overall goal of improving this manuscript. Wordings such as "overly fixated" do not contribute to a constructive discussion.
>
> Due to the limited constructiveness of the discussion so far, I am considering lowering my score.

---

> > ### Author Response · Authors · 2024-11-25
> >
> > Thank you for your feedback. The term "fixated" was indeed inappropriate and lacked composure. We apologize and have corrected it. We sincerely appreciate the time reviewers have taken for constructive discussion, especially within a limited timeframe. As we believe the contributions of this study are significant, we kindly ask for your continued engagement for a little while longer.

---

### Official Review · Reviewer_Vqs4 · 2024-11-12

**Soundness:** 3
**Presentation:** 4
**Contribution:** 4
**Rating:** 6
**Confidence:** 3

**Summary:**

This work derives a closed-form ridgelet transform for a specific class of neural networks, that is defined as joint-group-equivariant networks. In this class of neural networks, the assumption is that not only input and output are equivariant to a group action (group-equivariant networks), but that instead also the model's parameters are equivariant to a representation of the same group action acting on the networks parameters. While this at first seems like a limitation of the presented theory, it provides insights to a broad range of models as demonstrated in the experiments

**Strengths:**

This work presents some important insights and most importantly a constructive universal approximation theorem for (joint) group equivariant neural networks. The presentation is clear, the proofs concise, and the introduction includes all necessary lemmas from group representation theory. In addition to the derivation of a ridgelet transform for joint group invariant networks, section 5 provides the corresponding ridgeless transform for n-layer fully connected neural networks.

**Weaknesses:**

First it appears to be a limitation of the work, that the presented theory only applies to joint group equivariant networks (that is, the same group action is applied to the network's parameters as to the input and output), instead of to the widely used group equivariant networks (in which no group action acts on the model parameters to reach equivariance). In Remark 1, also Figure 1, it is however stated that the group-equivariant networks are included as a subclass of joint equivariant networks, as those for which G acts trivially on the parameter domain.

This raises some concerns, if all of the developed theory applies to this case. Theorem 4, which is the core of the established theory and defines the ridgelet transform, relies on irreducible representations of the group actions, both for the input as for the parameter space.
If however we would require the irreducible trivial representation for the parameter space but not for the input space then the question arises if all of the proofs still hold for this case. This might trivially be the case but because of its importance a comment on this special case would strengthen the manuscript.

Further information on the underlying assumptions for the non-linearities is needed. For group equivariant networks it is well-known that arbitrary non-linearities might destroy group-equivariance. It would be important to state which assumptions need to hold for the non-linearities for the joint group equivariant case and group equivariant case (in which G acts trivially on the parameter domain).

**Questions:**

- Does the derivation completely hold for the case in which the irreducible trivial representation is required for the parameter domain to include the group-invariant case?

- Are there any specific underlying assumptions for the non-linearities and how does this relate to the special requirements that non-linearities usually need to fulfill to preserve group equivariance?

---

> ### Author Response · Authors · 2024-11-18
>
> We thank the reviewer for their positive score and productive questions.
>
> ## Comments on Weaknesses
>
> - First it appears to be a limitation of the work, that the presented theory only applies to joint group equivariant networks... In Remark 1, also Figure 1, it is however stated that the group-equivariant networks are included as a subclass of joint equivariant networks, as those for which G acts trivially on the parameter domain.
>
> Certainly, joint-G-equivariance is not a restriction but an extension of the classical notion of G-equivariance, and in fact, all G-equivariant feature maps are joint-G-equivariant. (We have emphasized this in Remark 1.)
>
> - This raises some concerns, if all of the developed theory applies to this case. Theorem 4, which is the core of the established theory and defines the ridgelet transform, relies on irreducible representations of the group actions, both for the input as for the parameter space. If however we would require the irreducible trivial representation for the parameter space but not for the input space then the question arises if all of the proofs still hold for this case. This might trivially be the case but because of its importance a comment on this special case would strengthen the manuscript.
>
> If we understand your concerns correctly, there seems to be a misrecognition in
>
> > irreducible representations of the group actions, **both** for the input as for the parameter space.
>
> In fact, the irreducibility is assumed only for $\pi$, the representation on the functions $f:X \to Y$ on data domain, but not for $\hat{\pi}$, the representation on the distribution $\gamma : \Xi \to \mathbb{R}$ on parameters. This asymmetry originates from the fact that our main theorem only focuses on (the universality of) $LM[\gamma;\phi]:X \to Y$, not on its dual $R[f;\psi]:\Xi \to \mathbb{R}$. However, when the dual $\hat{\pi}$ is irreducible, then we can state $R \circ LM[\gamma] = \gamma$ for any $\gamma \in L^2(\Xi)$ (the order of composition is reverted from $LM \circ R[f] = f$). (We have supplemented this in Remark 3-2.)
>
> - Further information on the underlying assumptions for the non-linearities is needed. For group equivariant networks it is well-known that arbitrary non-linearities might destroy group-equivariance. It would be important to state which assumptions need to hold for the non-linearities for the joint group equivariant case and group equivariant case (in which G acts trivially on the parameter domain).
>
> We agree that this is an important direction of research in deep learning theory. Both necessary and sufficient conditions on activation functions for universality remain open problems. As you suggested, nonlinearity can easily break the classical (i.e. non-joint) equivariance. On the other hand, joint-equivariance is more likely to hold. In fact, in fully-connected networks (Sec.5) and quadratic-form networks (Sec.6), the joint-equivariance holds for any activation function. We note that joint-equivariance does not immediately imply universality, that is, the constant $((\phi,\psi))$ in Theorem 4 could degenerate to zero or diverge depending on the feature maps. For the case of depth-2 fully-connected networks, it is known that the constant  is zero if and only if the activation function is a polynomial function. To this date, however, such a condition is shown in a case-by-case manner, and the general principle is not known. In general, such a condition can be investigated in a case-by-case manner. Fortunately, we can use the closed-form expression of the ridgelet transform to our advantage. (We have  supplemented this in Remark 3-3.)
>
> ## Answers to Questions
> Please refer to Comments on Weaknesses.

---

> > ### Author Response · Authors · 2024-11-24
> >
> > As the discussion period is coming to an end, we would like to kindly remind you that we have addressed both of the questions raised by the reviewers. We understand you are busy, but we would greatly appreciate your cooperation in revising the score accordingly. Thank you for your consideration.

---

> ### Comment · Reviewer_Vqs4 · 2024-11-24
>
> *Regarding irreducible representations:*
>
> I thank the authors for their comments. I would suggest that a discussion of above points is being added to the paper as well as I believe that the "misrecognition" regarding irreducible representations could be prevented by adding an explicit discussion of this fact to the main body of the paper.
>
> *Regarding G-equivariance:*
>
> Further, I suggest that there should be a dedicated paragraph in the paper which lays out the theory for the special case of (non-joint) G-equivariant networks, as this is one of the main generalizations of this work with respect to previous work.
>
> Further comments with respect to nonlinearities are needed in this context. It is not clear under which assumptions the developed theory still holds for the (non-joint) G-equivariant case in the presence of nonlinearities and the necessary assumptions need to be clearly stated. Remark 1 states that G-equivariance is a special case of joint-G-equivariance; this however seems to contradict the following statement of the authors in this rebuttal: "nonlinearity can easily break the classical (i.e. non-joint) equivariance. On the other hand, joint-equivariance is more likely to hold." If the theory holds for joint-G-equivariance I would assume that it also should hold for the special case of G-equivariance without restriction. Also the statement that joint-equivariance is "more likely to hold" is not well defined. I encourage the authors to discuss this aspect more clearly.
>
> In its current form, the manuscript does not develop above points sufficiently and the discussion in the rebuttal has not yet completely answered my concerns. A dedicated paragraph that discusses the extent to which the developed theory holds for G-equivariant networks is needed for clarification and would greatly improve the manuscript.
>
> I appreciate that the authors added section 7 with an example for a G-equivariant network. This section however does not explain how the current framework of this paper could be applied to G-equivariance, but instead refers to the previously existing framework in Sonoda et al. (2022a). This does not shed light on the applicability of the method presented here.
>
> Further, there seem to be some references in section 7 to equations and sections in Sonoda et al. (2022a), which need to be made more specific:
>
> "The feature map is defined in Eq.19 as"
>
> "As showcased in Section 5, ..."
>
> I assume these refer to Eq. 19 in Sonoda et al. (2022a), and section 5 in Sonoda et al. (2022a). However this is just an assumption and should be explicitely stated in the text.
>
> Applicability to G-equivariant networks of the theory developed in this paper, in conjunction with the exact assumptions that need to be fulfilled, would be a major contribution of this work, that should not only be mentioned in Remark 1, and reduced to a reference to Sonoda et al. (2022a), but instead clearly worked out in further detail here.

---

> > ### Author Response · Authors · 2024-11-25
> >
> > We appreciate the reviewer’s comment.
> >
> > > Regarding irreducible representations: … I would suggest that a discussion of above points is being added to the paper as well as I believe that the "misrecognition" regarding irreducible representations could be prevented by adding an explicit discussion of this fact to the main body of the paper.
> >
> > This has been clarified on page 5 soon after their introduction and Remark 3 of the revised manuscript. We would appreciate it if you could kindly review it.
> >
> > > Further, I suggest that there should be a dedicated paragraph in the paper which lays out the theory for the special case of (non-joint) G-equivariant networks, as this is one of the main generalizations of this work with respect to previous work.
> >
> > This has been added to Section 7. We would appreciate it if you could review it.
> >
> > > Further comments with respect to nonlinearities are needed in this context. It is not clear under which assumptions the developed theory still holds for the (non-joint) G-equivariant case in the presence of nonlinearities and the necessary assumptions need to be clearly stated.
> >
> > > Also the statement that joint-equivariance is "more likely to hold" is not well defined. I encourage the authors to discuss this aspect more clearly.
> >
> > > In its current form, the manuscript does not develop above points sufficiently and the discussion in the rebuttal has not yet completely answered my concerns. A dedicated paragraph that discusses the extent to which the developed theory holds for G-equivariant networks is needed for clarification and would greatly improve the manuscript.
> >
> > It is mathematically challenging to clearly state the necessary and/or sufficient conditions under which group equivariance is preserved. This would likely involve theorems in representation theory, which are beyond the scope of this study. Therefore, in this work, we limit ourselves to providing specific examples.
> >
> > First, the examples are clear. Please take a closer look at the examples in Sections 5 (and 6). In the feature map $\sigma(a^\top x-b)$, applying the affine group $G = (L,t) \in GL(m) \ltimes \mathbb{R}^m$ to $x$ alone does not make it group-equivariant (i.e., $\sigma(a^\top Lx - b) \neq L \sigma(a^\top x -b) + t$). However, by appropriately applying $G$ to parameter $(a,b)$ as well, the entire expression $a^\top x−b$ can be made group-invariant. In this case, $\sigma(a^\top x - b)$ remains group-invariant regardless of $\sigma$. Such dual actions are relatively easy to find when a specific expression like $\phi(x, (a,b)) = \sigma(a^\top x−b)$ is given—one simply needs to solve a system of equations to make the entire expression group invariant.
> >
> > Next, let us address the difficulty of the general case. First, in case (1) where the expressions for parameters can be freely determined, joint group-equivariant mappings can always be constructed, as shown in Lemma 1. However, in case (2) where specific expressions for parameters are given in advance, determining the conditions under which a group action exists that satisfies joint group equivariance becomes a challenging problem.
> >
> > Figure 2 provides a geometric interpretation of a joint equivariant map. Following Taco Cohen, the parameter domain is the base (blue), the feature domain is the fiber (black), and the feature map is a section (vector field, orange) on the $G$-bundle. In case (a), the problem involves identifying the blue space and the group action whose orbit corresponds to the orange curve, given the black line space and its group action. Lemma 1 shows that such an action can always be found by letting the base as group $G$ itself. On the other hand, case (b) involves determining the group action that results in the orange graph being an orbit, given the black and blue spaces and the orange graph itself. The problem of describing the existence conditions for such a group-equivariant map is geometrically complicated and thus more difficult to address.
> >
> > > Remark 1 states that G-equivariance is a special case of joint-G-equivariance; this however seems to contradict the following statement of the authors in this rebuttal: "nonlinearity can easily break the classical (i.e. non-joint) equivariance. On the other hand, joint-equivariance is more likely to hold." If the theory holds for joint-G-equivariance I would assume that it also should hold for the special case of G-equivariance without restriction.
> >
> > There is no contradiction here, because joint equivariance is a more relaxed condition than equivariance. (Thus, the following reasoning is false: "If the theory holds for joint-equivariance I would assume that it also should hold for the special case of equivariance without restriction.")
> >
> > (contd)

---

> > > ### Author Response · Authors · 2024-11-25
> > >
> > > The statement "if equivariance, then joint equivariance" is true because joint equivariance with a trivial action on the parameters is (non-joint) equivariance. However, the converse is not always true. In fact, consider the specific example where $\sigma = \tanh$ and the feature map is $\sigma(ax−b)$. This is joint-equivariant but not equivariant.
> > >
> > > > I appreciate that the authors added section 7 with an example for a G-equivariant network. This section however does not explain how the current framework of this paper could be applied to G-equivariance, ...
> > >
> > > We will supplement more explanations not (only) in Remark 1 but in Section 7. Please note that we will not be able to respond until Tuesday due to our work schedule.

---

> > > > ### Comment · Reviewer_Vqs4 · 2024-11-27
> > > >
> > > > 'There is no contradiction here, because joint equivariance is a more relaxed condition than equivariance. (Thus, the following reasoning is false: "If the theory holds for joint-equivariance I would assume that it also should hold for the special case of equivariance without restriction.")'
> > > >
> > > > I see the point, perhaps some of these aspects could be added to the main manuscript to avoid any possible confusion.

---

> > > > > ### Author Response · Authors · 2024-11-27
> > > > >
> > > > > > I appreciate that the authors added section 7 with an example for a G-equivariant network. This section however does not explain how the current framework of this paper could be applied to G-equivariance, ...
> > > > >
> > > > > We appreciate your productive suggestions. We have major updated Section 7 and moved it to Appendix B.
> > > > > Below is an excerpt from the comments for all:
> > > > >
> > > > > > We have presented **additional novel result**: the ridgelet transform for depth-$n$ **group convolutional networks (GCNs)**.
> > > > > While depth-$n$ fully-connected network (FCN) is the most typical example of joint-equivariant (but not equivariant) learning machine, depth-$n$ GCN is the most typical example of equivariant (and thus joint-equivariant) learning machine. The paper is now well balanced with these **two key examples**.

---

### Author Response · Authors · 2024-11-20

We appreciate the reviewers' detailed feedback and constructive criticism.
We have revised our manuscript as follows:
- moved some proofs to Appendix A
- supplemented comments and remarks (highlighted in cyan), especially
  - Remark 2 and Figure 2: Interpretations of joint-equivariant map
  - Remarks 3 and 4: Comments on main theorem
  - ~~Section 7: Example for Classical Equivariant Network~~
    - moved to Appendix B: Example: Depth-$n$ Group Convolutional Network

---

> ### Author Response · Authors · 2024-11-27
>
> According to the suggestions from Reviewer Vqs4, we have major updated "Section 7: Example for Classical Equivariant Network", and presented **additional novel result**: the ridgelet transform for depth-$n$ **group convolutional networks (GCNs)**.
> While depth-$n$ fully-connected network (FCN) is the most typical example of joint-equivariant (but not equivariant) learning machine,
> depth-$n$ GCN is the most typical example of equivariant (and thus joint-equivariant) learning machine. The paper is now well balanced with these **two key examples**.
> Because the section gets longer, we have moved it (from Section 7) to
> - Appendix B: Example: Depth-$n$ Group Convolutional Network

---

### Meta-Review · Area_Chair_SHWb · 2024-12-23

**Metareview:**

This paper presents a constructive universal approximation theorem for learning machines equipped with joint-group-equivariant feature maps, utilizing group representation theory. The authors introduce the concept of joint-group-equivariance, which generalizes classical group-equivariance and includes fully-connected networks. The main theorem aims to unify universal approximation theorems for both shallow and deep networks, providing a closed-form expression for parameter distribution via the ridgelet transform. The reviewers acknowledged the novel theoretical contributions of this paper and the validity of the results. However, a concern was raised about the similarity of this paper and [1], in terms of the presentation of the results and the proof techniques. This concern was not resolved after a lengthy discussion process.

[1] S. Sonoda et al., Joint Group Invariant Functions on Data-Parameter Domain Induce Universal Neural Networks. In Proceedings of the 2nd NeurIPS, Workshop on Symmetry and Geometry in Neural Representations, Proceedings of Machine Learning Research, PMLR, 2024.

**Additional Comments On Reviewer Discussion:**

Reviewer oWLq pointed out a major concern about the similarity of this paper and a previous paper in terms of results and techniques. The reviewer did acknowledge the novelty of this paper's main result, but believed that the technical contributions were insufficient for publication given the high overlap. After several rounds of communication, the reviewer's concern remained.

---

### Decision · Program_Chairs · 2025-01-22

Reject